# On CA1 ripple oscillations in rats and the reassessment of asynchronicity evidence

Robson Scheffer-Teixeira[1]*, Adriano BL Tort[2]

[1]Department of Psychology, Federal University of Paraiba, João Pessoa, Brazil; [2]Brain Institute, Federal University of Rio Grande do Norte, Natal, Brazil

## eLife Assessment

This **important** study provides new insights into the synchronization of ripple oscillations in the hippocampus, both within and across hemispheres. Using carefully designed statistical methods, it presents **compelling** evidence that synchrony is significantly higher within a hemisphere than across. This study will be of interest to neuroscientists studying the hippocampus and memory.

*For correspondence:
schefferteixeira@gmail.com

Competing interest: The authors declare that no competing interests exist.

**Abstract** Sharp-wave ripples (SWRs) are hippocampal network oscillations associated with memory consolidation. They are characterized by the co-occurrence of fast and slow field potentials across CA1 layers: the fast-frequency oscillations, known as ripples, are prominent in the pyramidal cell layer, where they coincide with increased neuronal spiking, while slower negative transients, referred to as sharp waves, occur simultaneously in the *stratum radiatum*. SWRs have traditionally been considered globally synchronous across the hippocampus; however, recent evidence suggests that ripples may be less synchronous than previously thought, particularly between the two hemispheres (Villalobos et al., 2017). In this study, we revisited this question using a unique dataset from probes spanning the septo-temporal axis of CA1 in rats. Our results demonstrate that ripples are phase-locked within but not between hemispheres, although their occurrence remains time-locked across both the septo-temporal axis and hemispheres. We also observed a similar synchronicity pattern for spiking activity: neurons are locally phase-coupled and globally time-coupled to ripple events. Interneurons exhibit a much stronger phase coupling to both ipsilateral and contralateral ripples than pyramidal neurons. These findings suggest that ripples are locally phase-coupled through pyramidal–interneuron interactions, with global time-locking likely driven by a common bilateral CA3 input and potentially modulated by interneuron circuits.

## Introduction

Rodents display a diverse behavioral repertoire, which is accompanied by distinct patterns of neuro-electrophysiological activity. During behaviors such as awake immobility, licking, chewing, face-washing, grooming, and slow-wave sleep (SWS), the rodent hippocampal local field potential (LFP) is dominated by a slow, non-rhythmic pattern called large irregular activity (LIA) (*Vanderwolf, 1969*; *Suzuki and Smith, 1987*). *O'Keefe, 1976* and later *O'Keefe and Nadel, 1978*, observed that, during LIA states, slow negative deflections occur in CA1 *stratum radiatum*, coinciding with the presence of high-frequency oscillations (100-250Hz) in the pyramidal cell layer. These coupled events are now collectively referred to as sharp-wave ripples (SWRs; see *Buzsáki, 2015*, for an extensive review). SWRs have been observed in several mammals, but predominately studied in rodents (*Buzsáki, 1989*; *Buzsáki, 2015*; *Buzsáki et al., 2003*; *Buzsáki et al., 1992*). Individual SWR events typically last between 40 and 100ms (*Buzsáki, 2015*), though new evidence suggests that even longer events (above 100ms) may correlate with increased cognitive load (*Fernández-Ruiz et al., 2019*).

It is widely accepted that CA1 SWRs initiate in the CA3 region as a result of the synchronous population firing of its recurrent pyramidal cells and local inhibitory network (*Tukker et al., 2013*). This activity propagates through the associational Schaffer collaterals and commissural pathway, ultimately leading to a sharp negative deflection of the field potential detected in the CA1 *radiatum* layer (*Buzsáki et al., 1983*; *Ylinen et al., 1995*; *Csicsvari et al., 1999a*; *Schlingloff et al., 2014*; *Csicsvari et al., 2000*). This sharp deflection occurs due to a large current sink caused by a transmembrane sodium intake by the apical dendrites of CA1 pyramidal neurons and marks the onset of an SWR event. During the sharp wave, a subset of CA1 pyramidal cells increases its firing rate, triggering a negative feedback mechanism through the activation of local interneurons, like parvalbumin-positive basket cells (PV+) and bistratified cells (*Klausberger et al., 2004*; *Klausberger and Somogyi, 2008*; *Klausberger, 2009*; *Csicsvari et al., 1999b*; *Varga et al., 2014*; *Somogyi and Klausberger, 2005*; *Ylinen et al., 1995*; *Chrobak and Buzsáki, 1996*; *Stark et al., 2014*). This CA1 pyramidal–interneuron network activity gives rise to fast, local field and membrane oscillations (five-cycle duration) in the CA1 *stratum pyramidale* (*Hulse et al., 2016*; *Stark et al., 2014*). These so-called ripples seem to be mostly driven by phasic inhibition (*Gan et al., 2017*) and electrical coupling via gap junctions (*Traub and Bibbig, 2000*; *Maier et al., 2003*). Both pyramidal cells and some subclasses of interneurons increase their firing rate during ripple events, phase-locking to the oscillation: pyramidal cells fire during the ripple trough, while interneurons fire early at the ensuing ascending phase (*Csicsvari et al., 1999b*; *Buzsáki et al., 2003*; *Lapray et al., 2012*). Although most studies have focused on SWR events occurring in septal (dorsal) CA1 (*Buzsáki, 2015*), ripples can be detected along the entire septo-temporal axis (*Patel et al., 2013*; *Nitzan et al., 2022*) and can propagate to downstream structures such as the *subiculum* and entorhinal cortex (*Chrobak and Buzsáki, 1994*; *Chrobak and Buzsáki, 1996*).

Since early evidence linking the hippocampus to memory (*Scoville and Milner, 1957*), SWRs have been widely studied over the decades, not only for their underlying mechanisms, but also for their potential involvement in synaptic plasticity and cognitive processes (*Buzsáki, 1986*; *Suzuki and Smith, 1987*; *Suzuki and Smith, 1988a*; *Ylinen et al., 1995*; *Maier et al., 2003*; *O'Neill et al., 2006*; *Schlingloff et al., 2014*; *Stark et al., 2014*; *Buzsáki, 1984*; *Girardeau and Zugaro, 2011*; *Roumis and Frank, 2015*; *Sadowski et al., 2016*; *Roux et al., 2017*; *Buzsáki, 2015*), particularly in relation to memory consolidation. Key findings in the literature include: (1) CA1 pyramidal cells form assemblies during ripple events, which are thought to underlie memory encoding and consolidation *O'Neill et al., 2006*; *Nakashiba et al., 2009*; *van de Ven et al., 2016*; *Taxidis et al., 2015*; (2) ripple disruption by electrical or optogenetic stimulation during awake behavior or SWS impairs performance in spatial tasks *Jadhav et al., 2012*; *van de Ven et al., 2016*; *Girardeau et al., 2009*; *Ego-Stengel and Wilson, 2010*; *Gridchyn et al., 2020*; (3) SWS ripple events increase and become more sustained after learning a new memory task *Eschenko et al., 2008*; *Ramadan et al., 2009*; (4) prolonging spontaneous SWRs through optogenetic stimulation improves performance in a spatial memory task *Fernández-Ruiz et al., 2019*; (5) CA1 pyramidal neurons active in SWR events during the encoding phase of a spatial task increase their co-firing in subsequent rest or SWS *O'Neill et al., 2006*; *van de Ven et al., 2016*; *Lopes-Dos-Santos et al., 2018*; (6) place cell sequences during ripples occur as preplay or replay, both in forward and reverse directions (*Carr et al., 2012*; *Diba and Buzsáki, 2007*; *Shin et al., 2019*), potentially supporting memory encoding and retrieval *Roux et al., 2017*; (7) CA1 SWRs are associated with prefrontal and cortical spindles, and with high-frequency oscillations in the lateral septum, which may promote hippocampal-cortical or -subcortical communication and long-term memory consolidation *Jadhav et al., 2016*; *Maingret et al., 2016*; *Tingley and Buzsáki, 2020*; and (8) increased ripple activity has been reported during the recollection of semantic and autobiographical memories in humans (*Norman et al., 2021*). In addition, SWR has been implicated in non-mnemonic functions, such as metabolic control via glucose regulation (*Tingley et al., 2021*; *Kaya et al., 2025*) and in sleep homeostasis (*Giri et al., 2024*; *Miyawaki and Diba, 2016*).

Despite the extensive literature on SWRs, fundamental aspects of their synchronicity within and between hemispheres have received comparatively less attention. Prior evidence suggests that SWRs in the CA1 area can emerge in both ipsilateral and contralateral synchrony (*Buzsáki, 1989*; *Buzsáki et al., 1992*; *Suzuki and Smith, 1987*; *Suzuki and Smith, 1988a*; *Patel et al., 2013*) or propagate along the hippocampal septo-temporal axis (*Patel et al., 2013*). Nevertheless, several claims of ripple synchrony across hemispheres remain underexplored or insufficiently replicated. More recently, *Villalobos et al., 2017* addressed this gap and brought evidence that most ripple events occur

asynchronously between hemispheres, challenging earlier findings. Specifically, they reported that ripple events were predominantly synchronized within ipsilateral recordings, with minimal synchrony detected contralaterally. Given the central importance of SWRs in memory processes reviewed above, here we aimed to clarify these conflicting results by performing a detailed quantitative analysis of intra- and inter-hemispheric ripple synchrony.

## Results

Due to a recent work suggesting that ripple oscillations occur asynchronously between hemispheres (*Villalobos et al., 2017*), here we sought to revisit their level of inter-hemispheric synchrony. We started by first certifying that we could reliably identify ripple events in electrodes placed at the *stratum pyramidale* of the hippocampal CA1 region. To that end, we (1) filtered LFP signals between 100 and 250 Hz, (2) detected ripple events in the top-most electrode of a given shank, and (3) performed a ripple event-triggered average for all unfiltered LFP signals of that shank (*Figure 1A*, left). The presence of fast field oscillations superimposed on a slower wave, positive at the top of the shank (closer to *stratum oriens*) and negative at the bottom (closer to *stratum radiatum*) confirmed the successful detection of ripple events (see *Figure 1B* for an example ripple event; *Chrobak and Buzsáki, 1994*; *Buzsáki et al., 1983*; *Suzuki and Smith, 1987*). To maximize detection accuracy, for each shank, we then (4) determined which electrode had the highest ripple amplitude (mean envelope over all ripple-triggered averages; *Figure 1A*, right), which marks the center of the pyramidal layer (*Mizuseki et al., 2011*). This analysis pipeline was further validated by visual inspection of individual SWR events across the entire shank. In the following subsections, the results were obtained using the electrode displaying the highest ripple amplitude. Notice that we will avoid referring to ripple events as SWRs since we did not use the negative sharp deflection as a detection criterion.

### Ripple features do not differ between hemispheres

Previous evidence suggests molecular, synaptic, and cognitive specializations between left and right hippocampi (*Shipton et al., 2014*; *Shinohara et al., 2008*), which could potentially manifest as asymmetric electrophysiological patterns. We thus verified basic features of ripple oscillations bilaterally recorded from the left and right CA1 regions. We did not find statistically significant differences in ripple abundance (*Figure 2A*; $right - left$ difference = 0.012, p-value=0.15), ripple mean peak frequency (*Figure 2C*; $right - left$ difference = −0.907, p-value=0.42), inter-ripple interval (*Figure 2E*; $right - left$ difference = −0.048, p-value=0.132), or the number of cycles per ripple event (*Figure 2G*; $right - left$ difference = 0.071, p-value=0.31). For all ripple features, the 95% confidence interval of effect size differences (right panels) crossed zero, indicating no statistically significant difference between left and right effect sizes (see *Figure 2* legend for detailed analysis). Moreover, the pooled distributions of ripple features were visually similar between the left and right hippocampi (*Figure 2B, D, F, H*). To further confirm these findings, we used the two One-Sided Tests (TOST) method to test for equivalence between ripple features in the left and right hippocampi. We found evidence for equivalence, under the specified equivalence range, for all ripple features: (1) ripple abundance (*bounds* = ±0.032, p-value=0.084), (2) ripple mean peak frequency (*bounds* = ±4.87, p-value=0.0076), (3) inter-ripple interval (*bounds* = ±0.12, p-value=0.012), and (4) number of cycles (*bounds* = ±0.28, p-value=0.044). Notice that the equivalence bounds are shown in the same raw units as the analyzed features.

### Ripples phase-lock within but not between hemispheres

Visual inspection of ripple-filtered signals suggested that ripple events occurred simultaneously across all shanks in the left and right hippocampi (*Figure 1D*; dashed vertical lines indicate detected ripples at shank 1). Moreover, we observed that infra-threshold oscillatory bursts, which were not detected as ripple events in shank 1, also appeared to occur simultaneously across shanks. We then proceeded to quantitatively measure phase synchronization between ripples recorded in the same (ipsilateral) and across hemispheres (contralateral) by means of phase–phase coupling (PPC).

To exemplify this analysis, *Figure 3A* shows a ripple event from one reference shank, along with the ripple-filtered signals from two other shanks: one ipsilateral and one contralateral to the reference. The instantaneous phase series was extracted and used to compute the circular distribution of phase differences (*Figure 3B*) and the associated PLV. While this example highlights a single ripple event,

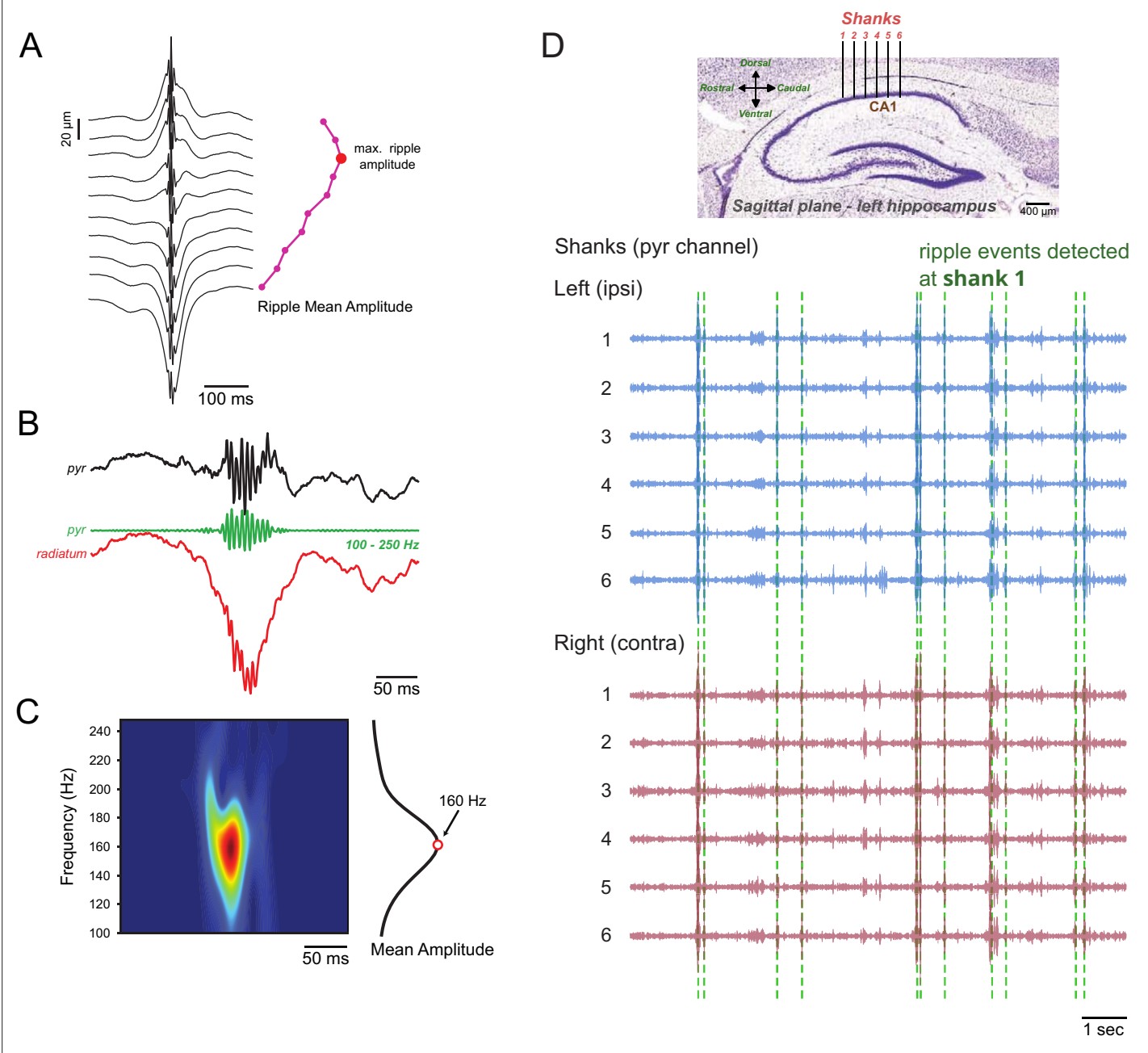

**Figure 1.** Tracking ripple events. (**A**) Example local field potential (LFP) averages aligned to ripple peaks during slow-wave sleep (left), and the mean ripple amplitude for all channels on a single shank (right). The ripple amplitude was obtained from the 100 to 250 Hz bandpass-filtered signal. For each shank, the electrode with the maximum amplitude (red dot) was selected for subsequent analyses. (**B**) Representative LFP traces during a sharp-wave ripple event. The raw signal from an electrode in the pyramidal layer is shown at the top, with the ripple-filtered signal in the middle. The bottom trace displays a raw LFP signal from the *stratum radiatum* layer, where the sharp wave manifests as a negative deflection associated with the ripple. (**C**) The spectrogram of a ripple event was obtained by wavelet transform (left). The center frequency (red dot) was defined as the frequency of maximum amplitude after averaging across the time window (right trace). (**D**) Top panel: Histology showing shanks placement scheme across dorsal CA1. Adapted from *Paxinos and Watson, 2006*. Middle and bottom panels: Representative ripple-filtered traces from several consecutive shanks. In this example, ripple events detected in shank 1 (marked by green dashed lines) were used as reference time points.

we analyzed the phase series from the entire signal (i.e., including outside ripple events) and from all combinations of shanks, as we will later compare these results with amplitude–amplitude coupling (AAC). We obtained similar results when restricting the analysis to detected ripple events (*Figure 3— figure supplement 1*).

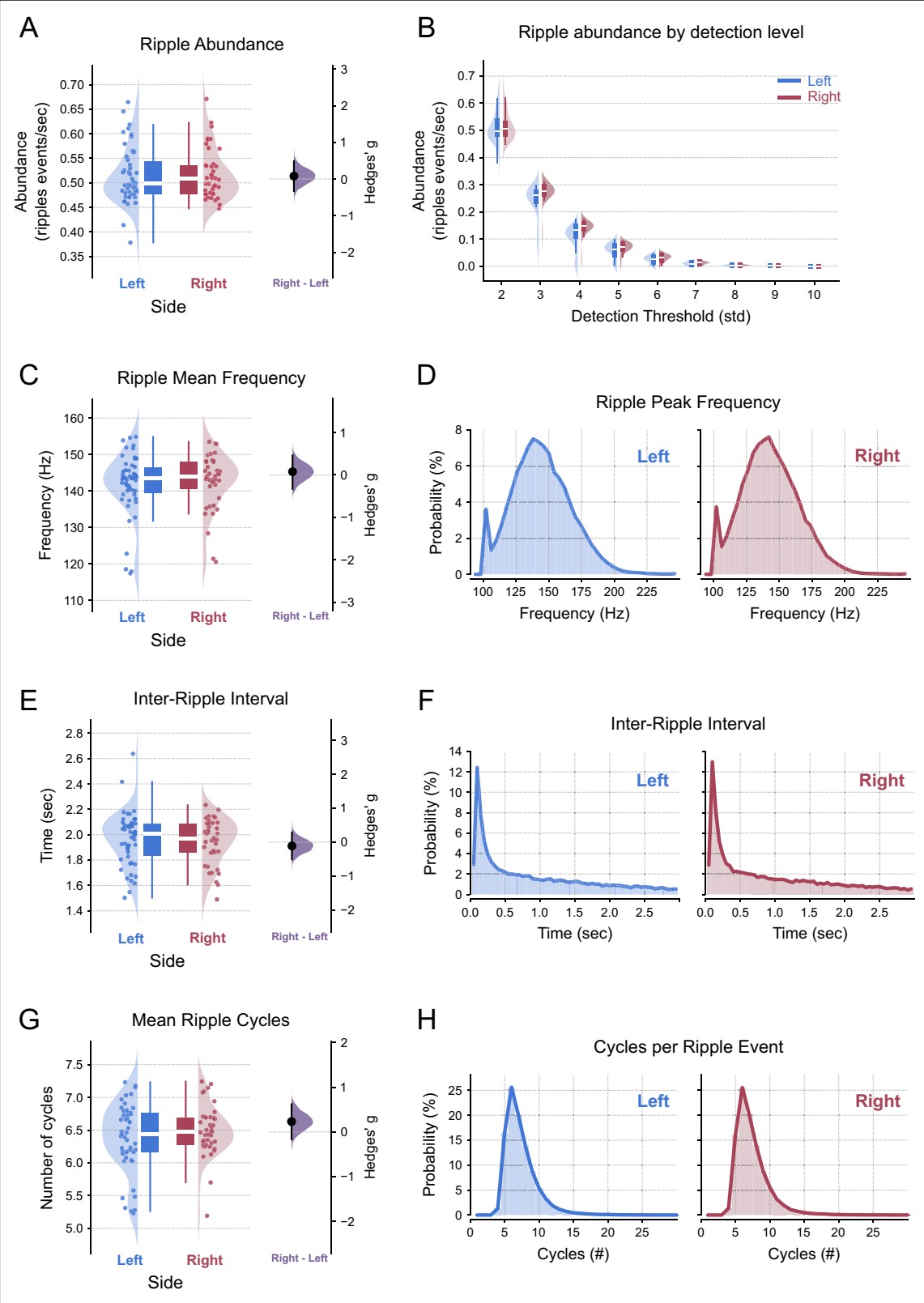

**Figure 2.** Ripple features do not differ between the left and right hippocampal hemispheres. (**A**) Ripple abundance. Left panel: Raincloud plot showing ripple events per second (abundance) in the left (blue) and right (red) hippocampus (LMMR, *side*, $N_{\text{left}} = 50$, $N_{\text{right}} = 40$, p-value=0.15). Right panel: Mean effect size (Hedges' *g*) for ripple abundance difference (permutation *t*- test, *right − left*, p-value=0.7). (**B**) Ripple abundance across increasing detection thresholds. Boxplots and half-violin plots illustrate ripple abundance in the left (blue) and right (red) hippocampus at multiple thresholds

*Figure 2 continued on next page*

*Figure 2 continued*

(LMMR, $N_{\text{left}} = 50$, $N_{\text{right}} = 40$, repeated measures across thresholds; *side* p-value=0.155; *threshold*: p-value=8.33×10$^{-177}$; *side vs. threshold* : p-value=0.27). (**C**) Ripple frequency. Left panel: Raincloud plots of ripple frequency (Hz) based on maximum amplitude in the left (blue) and right (red) hippocampus (LMMR, *side*, $N_{\text{left}} = 50$, $N_{\text{right}} = 40$, p-value=0.42). Right panel: Mean effect size (Hedges' *g*) for inter-ripple interval difference (permutation *t*-test, *right − left*, p-value=0.73). (**D**) Distribution of the pooled ripple frequency in the left (blue) and right (red) hippocampus. (**E**) Inter-ripple interval. Left panel: Raincloud plots showing the time intervals (in seconds) between consecutive ripple events in the left (blue) and right (red) hippocampus (LMMR, *side*, $N_{\text{left}} = 50$, $N_{\text{right}} = 40$, p-value=0.13). Right panel: Mean effect size (Hedges' *g*) for the inter-ripple interval difference (permutation *t*-test, *right − left*, p-value=0.6). (**F**) Distribution of the pooled inter-ripple intervals in the left (blue) and right (red) hippocampus. (**G**) Number of ripple cycles. Left panel: Raincloud plots displaying the average number of cycles per ripple event in the left (blue) and right (red) hippocampus (LMMR, *side*, $N_{\text{left}} = 50$, $N_{\text{right}} = 40$, p-value=0.31). Right panel: Mean effect size (Hedges' *g*) for ripple cycles difference (permutation *t*-test, *right − left*, p-value=0.28). (**H**) Distribution of the pooled mean ripple cycle count in the left (blue) and right (red) hippocampus. In the left panels of A, C, D, and E, the dots represent the mean value per shank pooled across all animals. In the right panels, the filled black circle indicates the mean difference, the purple half-violin plot displays the distribution of 5000 bootstrapped mean differences, and the vertical line around the mean shows the bootstrap 95% confidence interval.

We found that ripples recorded from ipsilateral shanks exhibited a higher PLV compared to those from contralateral shanks (*Figure 3C*; *contra − ipsi* difference = -0.445, p-value≈0), a result consistent with previous studies (*Ylinen et al., 1995*; *Chrobak and Buzsáki, 1996*; *Patel et al., 2013*). Specifically, contralateral shanks displayed a PLV reduction of 84.6% relative to ipsilateral ones. We also observed that PLV levels were correlated with shank distance (*Figure 3D*). Ipsilateral phase synchronization decayed at a rate of -0.076 per 200 μm, corresponding to a 10.73% reduction relative to the intercept. In contrast, for contralateral PLV across heterotopic sites, we observed a much lower decay rate of -0.001 per 200 μm (*Figure 3D*), which corresponds to only 1.56% decay relative to the intercept.

We further studied phase synchronization by using an alternative approach based on cross-correlation of ripple-filtered LFPs. In this analysis, we used the cross-correlation peak value at zero-lag as the phase synchronization metric (*Figure 3E*). Consistent with the previous results, ripples from ipsilateral shanks had a much higher cross-correlation amplitude at zero lag than those from contralateral shanks (*Figure 3F*; *contra − ipsi* difference = -2.7×10$^6$, p-value≈0), corresponding to a 89.4% reduction in contralateral relative to ipsilateral shanks. When classifying by shank distances, ipsilateral shanks showed a negative decay of phase synchronization at a rate of -0.41×10$^6$ per 200 μm, which corresponds to a 10.23% reduction relative to the intercept. For contralateral correlation across heterotopic sites, we observed a much lower rate of -0.0094×10$^6$ per 200 μm, corresponding to a 2.44% reduction relative to the intercept.

We conclude that CA1 ripple oscillations within the same hemisphere are strongly phase-locked over short distances, while ripples from different hemispheres exhibit weak coherence at the phase level.

## Ripple events are synchronous at the amplitude level

The lower inter-hemispheric phase synchrony does not imply that ripple events do not co-occur. In fact, as previously mentioned, visual inspection of ripple activity indicated that burst events occur with some level of global synchrony (*Figure 1D*). We next quantitatively investigated whether ripple events are synchronous between hemispheres at the amplitude level, by assessing whether their instantaneous amplitudes co-vary. To quantify the degree of amplitude coupling, we employed the Pearson correlation coefficient *r* (*Figure 4A, B*).

At the group level, we found a statistically significant correlation between the amplitude of ripples recorded from ipsilateral and contralateral shanks (*Figure 4C*, *contra − ipsi* difference = -0.099, p-value=9.5×10$^{-44}$). While the correlation coefficients for contralateral shanks were significantly lower than those for ipsilateral shanks, their absolute values still fell within a high correlation range, showing only an 11.2% reduction relative to ipsilateral shanks. We also found that amplitude coupling correlates with shank distance (*Figure 4D*). Specifically, ipsilateral amplitude coupling decayed at a rate of -0.028 per 200 μm, corresponding to a 3% reduction relative to the intercept, while contralateral amplitude coupling decayed at a lower rate of -0.010 per 200 μm, corresponding to a 1.3% reduction relative to the intercept.

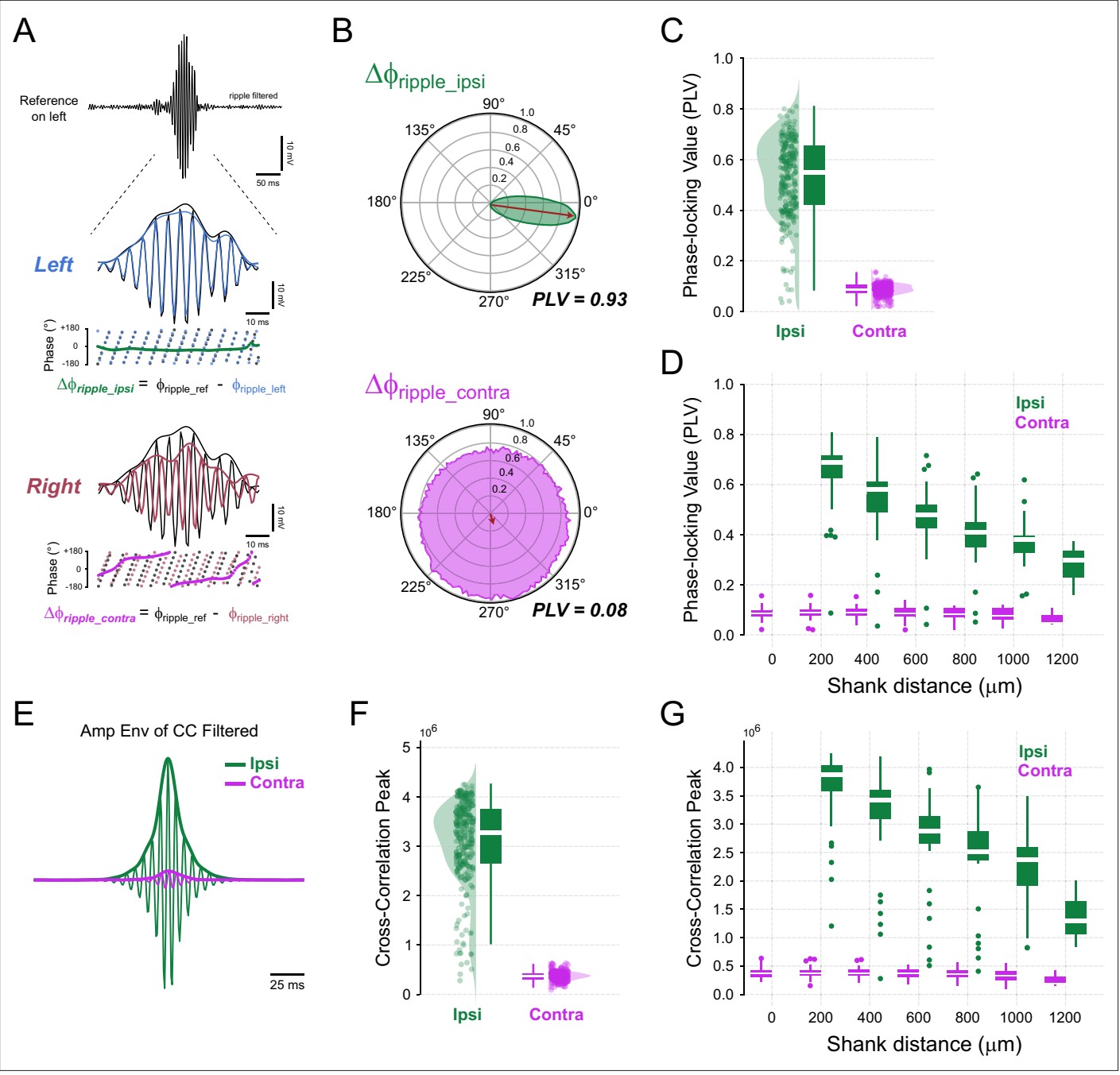

**Figure 3.** Ripple-filtered signals are phase-locked within but not between hemispheres. (**A**) Top: A representative reference ripple event (100–250 Hz; black). Middle and bottom: The ripple event on an ipsilateral shank (left hemisphere, blue) exhibits constant phase difference (green), whereas the ripple event on the contralateral hippocampus (right hemisphere, red) shows a non-constant phase difference (pink) to the reference event. (**B**) Polar histograms displaying phase differences between ripples in the reference shank and ipsilateral (top) or contralateral (bottom) shank. Phase-locking value (PLV) denotes the inter-regional phase coupling metric, defined as the length of the mean resultant from unitary vectors (red arrows). (**C**) Raincloud plots showing PLV for ipsilateral and contralateral shanks (LMMR, *side*, $N_{ipsi} = 220$, $N_{contra} = 245$, p-value≈0). (**D**) Same as (**C**), with data sorted by inter-shank distances. The regression line slope for ipsilateral is –0.076 for 200 µm inter-shank distance (LMMR, $N_{ipsi} = 220$, p-value=$4.94\times10^{-78}$). Regression line slope for contralateral is –0.001 for 200 µm inter-shank distance (LMMR, $N_{contra} = 245$, p-value=0.053). (**E**) Thin traces show a representative cross-correlation between the ripple-filtered signal from the reference shank and the ripple-filtered signal from ipsilateral (green) or contralateral (pink) shanks. The thick traces depict the amplitude envelopes of these cross-correlations; the peak amplitude is taken as a phase coupling metric. (**F**) Raincloud plots of cross-correlation maximum peak of ripple-filtered signal for ipsilateral and contralateral shanks (LMMR, *side*, $N_{ipsi} = 220$, $N_{contra} = 245$, p-value≈0). (**G**) Same as (**F**), but data sorted by inter-shank distances. Regression line slope for ipsilateral is -0.41×10⁶ (cross-correlation peak) for 200 µm inter-shank

*Figure 3 continued on next page*

*Figure 3 continued*

distance (LMMR, $N_{\text{ipsi}} = 220$, p-value=4.03×10⁻⁴⁶). Regression line slope for contralateral is –0.094×10⁶ (cross-correlation peak) for 200 μm inter-shank distance (LMMR, $N_{\text{contra}} = 245$, p-value=0.006).

The online version of this article includes the following figure supplement(s) for figure 3:

**Figure supplement 1.** Ripple events are phase-locked within but not between hemispheres.

Finally, we examined ripple event synchronization through the cross-correlation of amplitude envelopes (*Figure 4E*). This analysis yielded similar results, showing a significant but slightly higher peak at zero lag for ipsilateral signals compared to contralateral ones (*Figure 4F*, *contra − ipsi* difference = -0.95×10⁶, p-value=9.9×10⁻¹⁸¹), corresponding to a contralateral reduction of 11.9% relative to ipsilateral. The cross-correlation peak also correlated with shank distance (*Figure 4G*). Ipsilateral peaks decayed at a rate of -0.22×10⁶ per 200 μm, corresponding to a 2.58% reduction relative to the intercept, while contralateral peaks across heterotopic sites decayed at a lower rate of -0.031×10⁶ per 200 μm, corresponding to a 0.44% reduction relative to the intercept.

We conclude that the amplitude coupling of CA1 ripple oscillations is higher within than between hemispheres and tends to decline with shank distance. Nevertheless, our results demonstrate generally high correlation coefficients, suggesting that ripples are globally time-locked.

## Phase coupling and amplitude coupling show different effect sizes

Thus far, we have observed stronger ipsilateral vs. contralateral effects for phase coupling compared to amplitude coupling metrics. As shown in *Figure 3C, F*, the distributions from ipsilateral and contralateral shanks exhibited greater overlap and similarity for amplitude coupling than for phase coupling metrics. This pattern was consistent when averaging within individual sessions (*Figure 4—figure supplement 1*) and also replicated in an independent dataset (*Figure 4—figure supplement 2*).

In summary, the relative percentage difference of *contra − ipsi* was smaller for amplitude coupling (11.2% for *r* and 11.9% for ripple amplitude cross-correlation) compared to phase coupling (84.6% for PLV and 89.4% for ripple-filtered cross-correlation). Furthermore, when analyzing only ipsilateral shanks, the decay rate (in *percent* of the intercept value) for amplitude coupling was slower (3% for *r* and 2.58% for ripple amplitude cross-correlation) compared to phase coupling (10.73% for PLV and 10.23% for ripple-filtered cross-correlation).

Therefore, to further quantify the differences observed between phase and amplitude couplings, we generated bootstrapped distributions of effect sizes (Hedges' *g*), which normalize the distributions and allow for comparison of distinct metrics. First, we confirmed that the ipsilateral-to-contralateral effect sizes for both phase (PLV) and amplitude couplings (*r*) were positive and significantly greater than zero, indicating a directional effect favoring higher values in ipsilateral shanks (*Figure 5A*, left). Moreover, when directly comparing the effect sizes for phase and amplitude coupling, we found a significantly higher effect for phase coupling (*Figure 5A*, right, Delta's *g*, permutation *t*-test, difference = 3.2, p-value≈0). This result was consistent when analyzing average effect sizes per session (*Figure 5B*, paired *t*-test, difference = 1.765, p-value=0.00254).

A similar pattern was also observed when comparing cross-correlation metrics, with a higher effect size for ripple-filtered signals than for ripple amplitude cross-correlations (*Figure 5C*, right, Delta's *g*, permutation *t*-test, difference = 3.28, p-value≈0). This finding was also consistent when averaging values per session (*Figure 5D*, paired *t*-test, difference = 2.757, p-value=0.0056).

These results suggest that ripples exhibit highly localized phase coupling that rapidly decays with shank distance, whereas amplitude coupling maintains a more global synchrony, resulting in smaller differences between ipsilateral and contralateral shanks, as well as across ipsilateral shank distances.

## Ripple phase and amplitude coupling are not influenced by novelty exposure

We investigated whether ripple synchronization changes after novel spatial learning. The number of ripple events increased following sessions involving novel maze exposure (*Figure 5—figure supplement 1A*, post–pre difference = 0.039, p-value = 6.85×10⁻¹²). Although the effect size of this increase was small, it was consistent across sessions (*Figure 5—figure supplement 1B*; paired *t*-test, post–pre difference = 0.038, p-value=6.6×10⁻⁴). Despite the higher ripple occurrence, we did not detect any

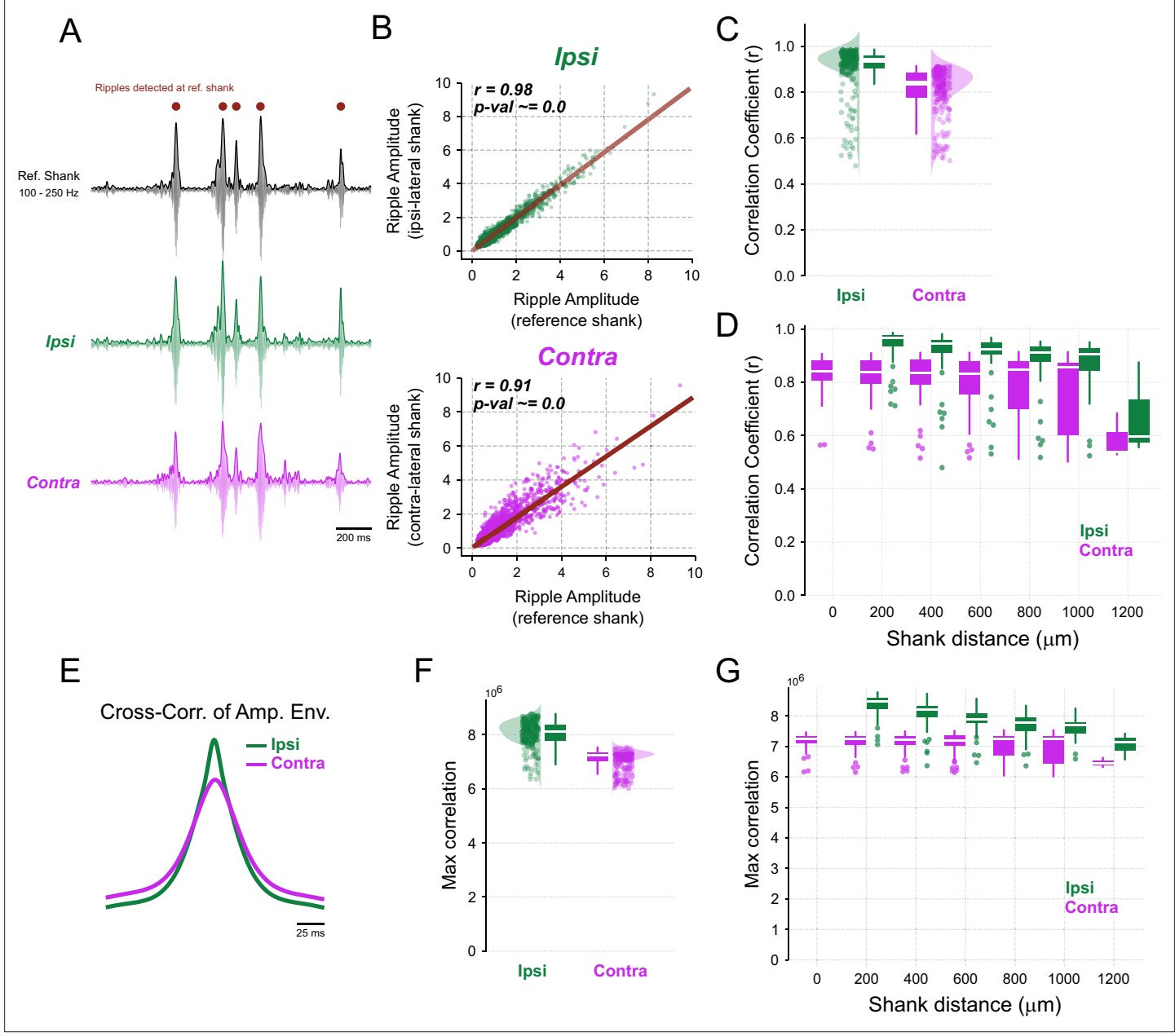

**Figure 4.** The amplitude of ripple events correlates within and between hemispheres. (**A**) Representative ripple-filtered signals (thin traces) and their instantaneous amplitude envelopes (thick traces). Dots represent detected ripple events at the reference shank. (**B**) Representative scatter plot of ripple amplitude during ripple events for the reference vs. ipsilateral shank (instantaneous amplitude values were averaged within 100 ms bins, 50 ms overlap). The Pearson's correlation coefficient ($r$) is used as a metric of amplitude coupling. (**C**) Raincloud plots showing correlation coefficients between the amplitude of ripple events for ipsilateral and contralateral shanks (LMMR, *side*, $N_{ipsi} = 220$, $N_{contra} = 245$, p-value=$9.5×10^{-44}$). (**D**) Same as (**C**), but data sorted by inter-shank distances. The regression line slope for ipsilateral is –0.028 for 200 µm inter-shank distance (LMMR, $N_{ipsi} = 220$, p-value=$1.7×10^{-14}$). Regression line slope for contralateral is –0.010 (Pearson's correlation coefficient) for 200 µm inter-shank distance (LMMR, $N_{contra} = 245$, p-value=$8.6×10^{-5}$). (**E**) Representative cross-correlation between the instantaneous ripple amplitude in the reference shank and in the ipsilateral (green) or contralateral (pink) shanks (the whole time series, and not only ripple events, was used in this analysis). (**F**) Raincloud plots of cross-correlation maximum peak of instantaneous ripple amplitude for ipsilateral and contralateral shanks (LMMR, *side*, $N_{ipsi} = 220$, $N_{contra} = 245$, p-value=$9.9×10^{-181}$). (**G**) Same as (**F**), but data sorted by inter-shank distances. Regression line slope for ipsilateral is -0.22×$10^6$ (cross-correlation peak) for 200 µm inter-shank distance (LMMR, $N_{ipsi} = 220$, p-value=$2.6×10^{-37}$). Regression line slope for contralateral is -0.031×$10^6$ (cross-correlation peak) for 200 µm inter-shank distance (LMMR, $N_{contra} = 245$, p-value=$3.5×10^{-4}$).

The online version of this article includes the following figure supplement(s) for figure 4:

**Figure supplement 1.** Single-session variability on ripple synchronization metrics.

**Figure supplement 2.** Ripple synchronizes between hemispheres at the amplitude but not phase level in an additional dataset.

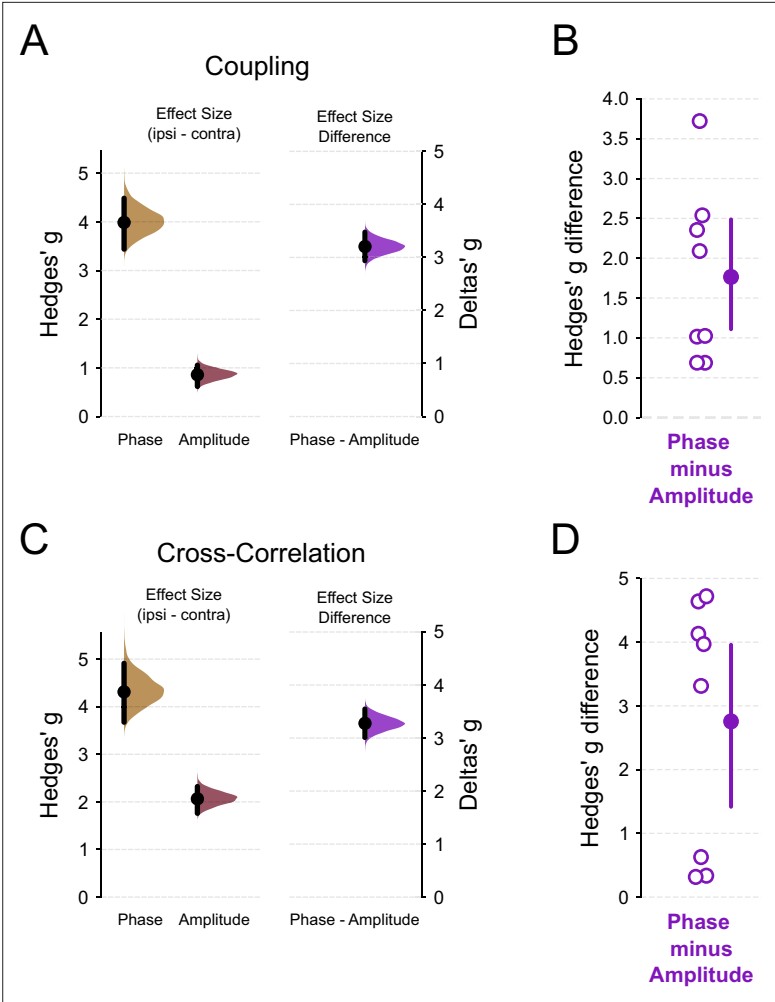

**Figure 5.** Ripple synchronizes between hemispheres at the amplitude but not phase level. (**A**) Left panel: Effect sizes for *ipsi − contra*, measured as Hedges' *g*, observed for phase coupling (*Figure 3C*) and amplitude coupling (*Figure 4C*). Positive values indicate that the mean coupling value is larger for ipsilateral shanks (permutation *t*-test, $N_{ipsi}$ = 220, $N_{contra}$ = 245, p-value≈0, p-value≈0). Right panel: Difference between the two effect size distributions shown on the left (permutation *t*-test, *phase − amplitude difference* = 3.2, p-value≈0). A value higher than zero indicates that the effect size for phase coupling is larger than for amplitude coupling. Data is presented as a half-violin plot for a 5000 bootstrap distribution of Hedges' *g* along with the mean and a bootstrap 95% confidence interval. (**B**) Mean distribution of the Hedges' *g* difference (*phase − amplitude*). Open purple circles represent the mean difference of shanks from the same recording session, that is, average of all combinations of ipsilateral and contralateral shanks (paired *t*-test, $N_{phase}$ = 8, $N_{amplitude}$ = 8, difference=1.765, p-value=0.00254). Values closer to zero indicate that phase and amplitude coupling metrics are approximately the same; on the other hand, values above zero indicate that the effect size of phase coupling is higher than amplitude coupling. Filled circle and vertical bar represent the mean and the bootstrap 95% confidence interval, respectively. (**C**) Same as in (**A**) but for cross-correlation analyses. Left panel: Effect sizes for *ipsi − contra*, measured as Hedges' *g* (permutation *t*-test, $N_{ipsi}$ = 220, $N_{contra}$ = 245, p-value≈0, p-value≈0). The phase label represents the cross-correlation maximum peak of ripple-filtered signal (*Figure 3F*); amplitude label, the cross-correlation maximum peak of instantaneous ripple amplitude (*Figure 4F*). Right panel: Difference between the two effect size distributions shown on the left (Deltas' *g*, *phase − amplitude* difference = 3.28, p-value≈0). (**D**) Same as (**B**), but for cross-correlation analyses (paired *t*-test, $N_{phase}$ = 8, $N_{amplitude}$ = 8, difference = 2.757, p-value=0.0056).

The online version of this article includes the following figure supplement(s) for figure 5:

**Figure supplement 1.** Experience-induced increase in ripple counts does not modify global synchrony.

differences in ripple phase or amplitude coupling between pre- and post-maze exposure (*Figure 5—figure supplement 1C–F*; post hoc pairwise *t*-tests showed no significant differences between post–pre comparisons for ipsilateral or contralateral metrics).

We next examined whether the type of track, linear (bidirectional running) or circular (unidirectional running), influenced inter-hemispheric coupling. In principle, linear track running could produce more symmetric sensory inputs to both hemispheres, as the two opposite running directions expose each hemisphere to similar visual and spatial cues. In contrast, circular track running might produce more asymmetric sensory inputs, with one hemisphere processing stimuli from the center of the track and the other from the surrounding walls. Nevertheless, as shown in *Figure 5—figure supplement 1*, our analysis did not reveal significant differences in inter-hemispheric coupling between animals running on linear vs. circular tracks.

## Ripple events are synchronous at larger time lags

Our work so far has demonstrated that ripple oscillations are phase-locked at ipsilateral sites and amplitude-locked across both ipsilateral and contralateral sites, suggesting that ripple events are globally time-locked across hemispheres. However, amplitude coupling serves only as a proxy for ripple co-occurrence. To directly quantify event coordination, we conducted further analyses to evaluate the timescale of event locking using ripple coincidence and ripple binned event cross-correlation metrics.

For ripple coincidence, which measures the average proportion of ripple events co-occurring within time windows of 5, 50, or 100 ms across all shank combinations, we observed higher coincidence proportions for ipsilateral shanks and longer time windows around ripple occurrences (*Figure 6A*; linear mixed model regression (LMMR), $side = -0.228$, p-value=$1.7\times10^{-102}$; $bin\,size = 3.2\times10^{-3}$, p-value=$9.4\times10^{-157}$; interaction = $1.4\times10^{-3}$, p-value=$1.2\times10^{-18}$). Importantly, the coincidence proportions in all conditions were substantially higher than the values reported by *Villalobos et al., 2017*, which reached up to 29% for ipsilateral shanks and up to 10% for contralateral shanks. Conversely, the effect size of the ipsi-to-contralateral proportion difference decreased with longer time windows, as the distributions of ipsilateral and contralateral coincidence proportions became more similar (*Figure 6B*; permutation *t*-test: 5 ms difference = 1.7, p-value≈0 difference = 0.769, p-value≈0 difference = 0.706, p-value≈0).

For ripple event cross-correlation, we analyzed ripple occurrence timestamps binned into time intervals of 5, 50, or 100 ms (*Figure 6C, D*). The cross-correlation value at zero lag was used as the synchrony measure (*Villalobos et al., 2017*). These bin widths correspond to the duration of a single ripple cycle (5ms) or an entire SWR event (50–100 ms) (*Chrobak and Buzsáki, 1996*). Similarly, we found higher cross-correlation counts for ipsilateral shanks and for longer bin windows (*Figure 6C*; LMMR, $side = -517$, p-value=$2.3\times10^{-135}$; $bin\,size = 5.196$, p-value=$1.4\times10^{-108}$; interaction = 2.94, p-value=$9\times10^{-20}$). The distribution of ipsi-to-contralateral counts became more similar as bin width increased, as reflected in the effect sizes of ipsi-to-contralateral differences, which were substantially larger for 5 ms compared to 50 and 100 ms bins (*Figure 6D*; permutation *t*-test, 5 ms difference = 1.92, p-value≈0 difference = 1.07, p-value≈0 difference = 0.957, p-value≈0).

In summary, ripple coincidence proportions and cross-correlation values for ripple counts binned at 5 ms were significantly higher for ipsilateral shanks, indicating stronger local synchronization at shorter timescales. However, for larger and biologically relevant time windows of 50 and 100 ms, ripple events exhibited broader global synchronization. These findings align with previous reports of global ripple synchrony at larger timescales (e.g., *Chrobak and Buzsáki, 1996*, see also their Figure 1E and F).

## Spiking activity phase-locks locally and time-locks globally to ripples

A last question in our study was how ipsilateral and contralateral ripples relate to spiking activity. To address this, we examined the phase and event coupling between spike times and ripples for pyramidal neurons and interneurons. A ripple-triggered firing rate analysis revealed an increase in firing rate during ripple events (*Figure 7A, B*). Interestingly, firing rates seemed modulated by ipsilateral ripple oscillations, with faster firing rate oscillations superimposed on a slower overall increase (see left panels of *Figure 7A, B*). In contrast, contralateral ripples only exhibited the overall firing rate increase, lacking the faster oscillatory firing rate modulation. Similarly, we observed a higher averaged spike-triggered ripple-filtered activity in ipsilateral shanks compared to contralateral ones. However,

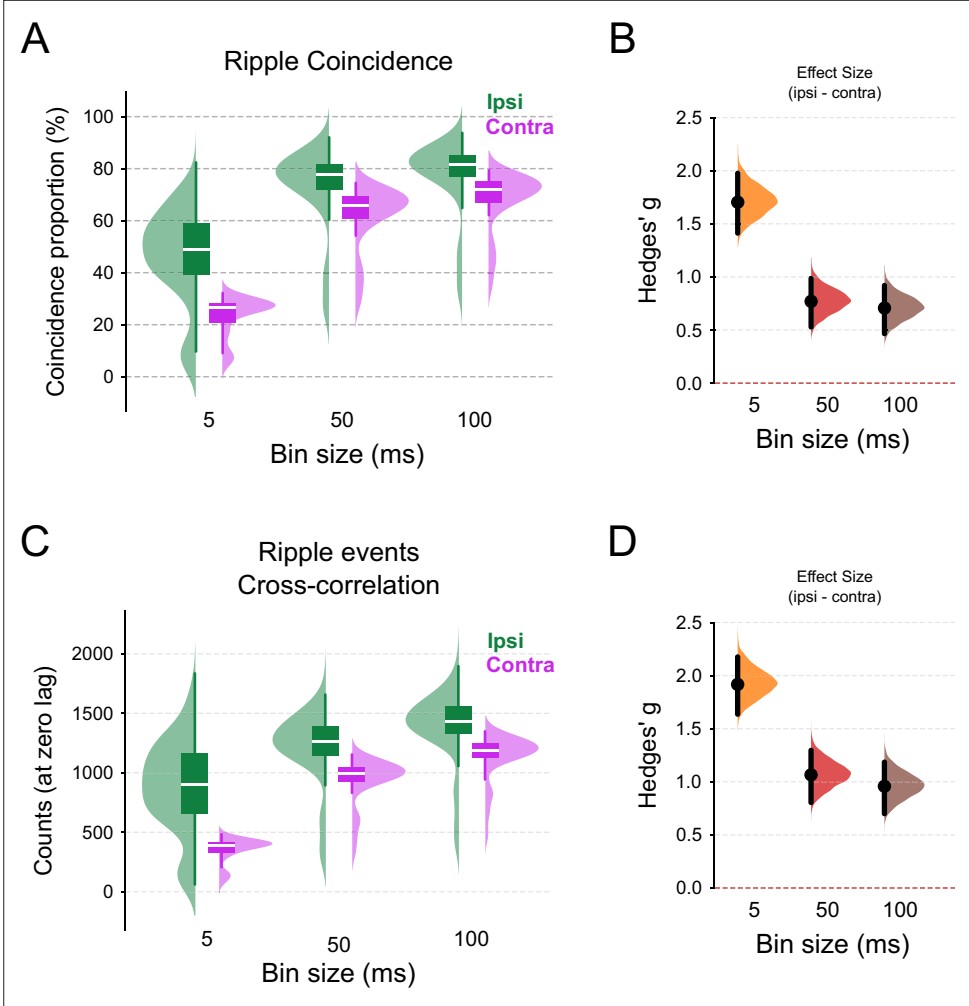

**Figure 6.** Ripple events occur in synchrony between hemispheres at the decisecond timescale. (**A**) Box and half-violin plots for ripple coincidence, defined as the percentage of overlapped ripple events in the given time windows (5, 50, or 100 ms). LMMR ($N_{ipsi} = 220$, $N_{contra} = 245$) revealed a main effect of the *side* variable (p-value=$1.2 \times 10^{-102}$) and *bin size* (p-value=$9.4 \times 10^{-157}$), and also an interaction effect of *side* vs. *bin size* size (p-value=$1.2 \times 10^{-18}$). Tukey HSD (FWER = 0.05) post hoc pairwise comparisons show significant differences between all conditions. Green represents ipsilateral events and pink contralateral ones. (**B**) Mean effect size (Hedges' *g*) for ripple coincidence differences across time bins (*ipsi − contra*; 5 ms difference = 1.7, p-value≈0 difference = 0.769, p-value≈0 difference = 0.706, p-value≈0). (**C**) Box and half-violin plots of peak values from cross-correlation of ripple events across 5, 50, or 100 ms time bins. In this analysis, ripples were coded as one if a single event occurred in that time bin, or zero otherwise. The cross-correlation value at zero lag was taken as the metric of synchrony. LMMR ($N_{ipsi} = 220$, $N_{contra} = 245$) revealed a main effect of the *side* variable (p-value=$2.3 \times 10^{-135}$) and *bin size* (p-value=$1.4 \times 10^{-108}$), and also an interaction effect of *side* vs. *bin size* (p-value=$9 \times 10^{-20}$). Tukey HSD (FWER = 0.05) post hoc pairwise comparisons show significant differences between all conditions, except '50 ms bin – contra' vs. '5 ms bin – ipsi'. (**D**) Mean effect size (Hedges' *g*) for cross-correlation peak of ripple events differences across time bins (*ipsi − contra*;

5 ms difference = 1.92, p-value≈0 difference = 1.07, p-value≈0 difference = 0.957, p-value≈0). For effect size figures B and D, a filled black circle indicates the mean difference, the purple half-violin plot displays the distribution of 5000 bootstrapped mean differences, and the vertical line around the mean shows the bootstrap 95% confidence interval. A red dashed line indicates the threshold where the effect size difference is not statistically significant.

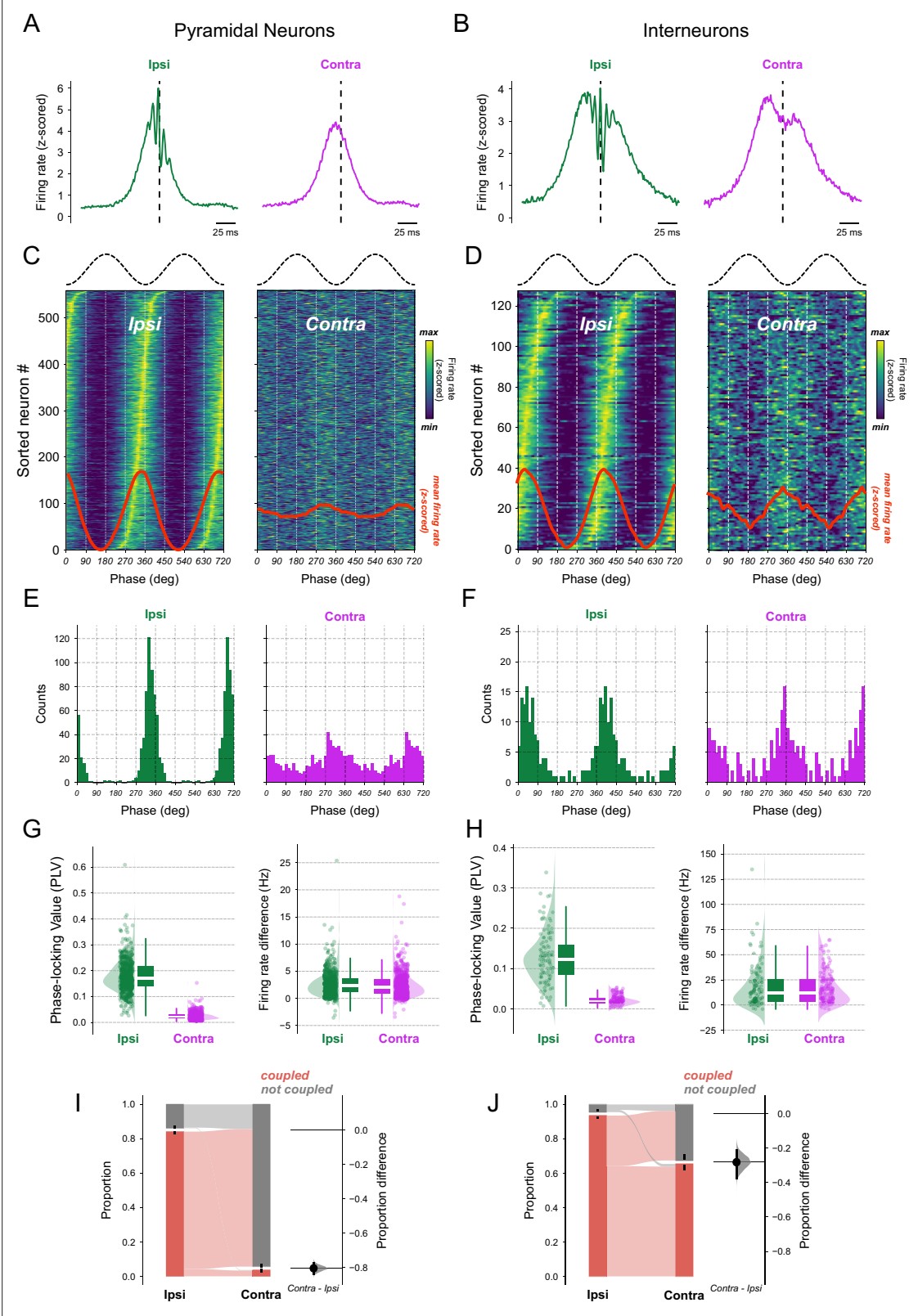

**Figure 7.** Spiking activity is globally coupled to ripple events. Ripple-triggered firing rate of ipsilateral (green) and contralateral (pink) CA1 pyramidal cells (**A**) or interneurons (**B**). Ripple peak amplitude was used as the trigger. Notice the presence of a ripple oscillatory component modulating firing rate in ipsilateral shanks. Spike-phase distributions for ipsilateral and contralateral ripple events for pyramidal (**C**) or interneurons (**D**). The top trace shows a ripple oscillation phase reference. In both the left and right 2D histograms, each line represents one neuron's z-scored firing rate, sorted by its ipsilateral

*Figure 7 continued on next page*

*Figure 7 continued*

peak ripple phase. The contralateral histogram preserves this ordering, enabling direct comparison across hemispheres. Red trace at the bottom shows the average firing rate for ripple phase across all neurons. (**E, F**) Histogram of the mean spike ripple-phase distribution from panels C and D for ipsi (green) and contra (pink) ripple events. Spike-ripple coupling metrics for pyramidal (**G**) or interneurons (**H**). Left panel: Spike coupling to ripple phase by means of phase-locking value (PLV). The PLV is calculated for each neuron individually. Right panel: Neuronal firing rate difference between inside and outside ripple events for ripples detected in ipsilateral and contralateral shanks. Positive values mean that the firing rate is higher during ripple. LMMR revealed an effect for *side* variable of pyramidal neurons phase coupling (G, left panel, $N_{pyr} = 559$), but not for firing rate difference (G right panel, $N_{pyr} = 559$). Likewise, LMMR revealed an effect for *side* variable in interneurons phase coupling (H, left panel, $N_{int} = 128$), but not for firing rate difference (H, right panel, $N_{int} = 128$). Left panel: Proportion of pyramidal neurons (**I**) or interneurons (**J**) significantly coupled to ipsilateral and contralateral ripple phase. Right panel: proportion difference of ripple phase-coupled neurons between contralateral and ipsilateral (pyramidal neurons, $N_{pyr} = 559$, *contra* − *ipsi* difference=-0.805, p-value≈0; interneurons, $N_{int} = 128$, *contra* − *ipsi* difference=-0.281, p-value≈0). The proportion difference measures the effect size between ipsi and contra. The filled black circle indicates the proportion difference, the gray half-violin plot displays the distribution of 5000 bootstrapped mean proportion differences, and the vertical line around the mean shows the bootstrap 95% confidence interval. Neuronal phase coupling was assessed by creating a PLV control distribution of 300 random time shifts of ±10 s in neurons timestamps relative to ripple phase series; neurons whose PLV was higher than the 95th percentile were deemed as significant.

The online version of this article includes the following figure supplement(s) for figure 7:

**Figure supplement 1.** Spikes are locally coupled to ripple signal but globally coupled to ripple amplitude across the septo-temporal axis.

**Figure supplement 2.** Spikes are locally phase-coupled to ripples and globally coupled to ripple events across the septo-temporal axis.

when considering ripple amplitude, the averaged spike-triggered ripple amplitude levels were similar for both sides (***Figure 7—figure supplement 1***).

To delve deeper, we examined the ripple-phase firing rate distribution. Neurons were circularly sorted by their ipsilateral peak firing phase, and this ordering was maintained in both ipsilateral and contralateral histograms to enable direct comparison (***Figure 7C, D***). Both neuron types showed strong phase modulation to ipsilateral ripples, with much smaller modulation to contralateral ripples. Moreover, pyramidal neurons exhibited ripple phase preference just before the ripple valley, whereas interneurons preferred phases just after the ripple valley (***Figure 7E, F***).

To quantify ipsilateral and contralateral ripple phase and event time-locking, we employed the PLV of spike phases (***Figure 7G, H***, left panels) and firing rate difference between inside and outside ripple events (***Figure 7G, H***, right panels). We found that the PLV for ipsilateral ripple phases was larger compared to contralateral ones (LMMR, pyramidal neurons, $N_{pyr} = 559$, coefficient = –0.155, p-value ≈ 0; interneurons, $N_{int} = 128$, coefficient = –0.106, p-value = $6.77 \times 10^{-85}$). On the other hand, we found no difference between the firing rate delta during ipsilateral and contralateral ripple events (LMMR, pyramidal neurons, $N_{pyr} = 559$, coefficient = –0.166, p-value = 0.218; interneurons, $N_{int} = 128$, coefficient = –1.191, p-value = 0.610).

Similar results were observed when analyzing PLV and firing rate differences across shank distances. Consistent with PPC between ipsilateral ripples, the spike phase coupling rapidly decayed with distance, suggesting that spike phase coupling is a local phenomenon (***Figure 7—figure supplement 2A, B***). Interestingly, pyramidal neurons exhibited a larger drop in PLV than interneurons from a local shank at 0 μm to a shank 200 μm. Namely, pyramidal neurons exhibited a drop of 0.27 in PLV, representing 59% of their PLV at 0 μm, while interneurons showed a smaller drop of 0.12, representing 38% of their PLV at 0 μm (t-test, $N_{pyr} = 526$, $N_{int} = 115$, mean difference = 0.16, p-value = $1.2 \times 10^{-29}$). In contrast, firing rate differences between inside and outside ripples remained constant across all inter-shank distances for both ipsilateral and contralateral shanks (***Figure 7—figure supplement 2C, D***).

Finally, we calculated the proportion of neurons significantly phase-coupled to ipsilateral and contralateral ripples using a control distribution of randomly shifted spikes (***Figure 7I, J***). A higher proportion of pyramidal neurons (85%) and interneurons (95%) were phase-coupled to local ipsilateral ripples. However, this proportion decreased for contralateral ripples ($N_{pyr} = 526$, proportion difference = –0.805, p-value ≈0, proportion difference = –0.281, p-value ≈0). Notably, contralateral ripple phase coupling was more prominent in interneurons than in pyramidal neurons (see also ***Figure 7—figure supplement 1C, D***).

In summary, CA1 pyramidal neurons and interneurons exhibit local phase coupling and global event locking to ripples. Additionally, interneurons display diminished but significant phase coupling to global ripples.

# Discussion

We have re-evaluated the level of inter-hemispheric synchrony of ripple oscillations. To our knowledge, this is the first study to fully quantify rodent CA1 ripple synchronization along the ipsilateral and contralateral longitudinal axes while controlling for bilateral homotopic and heterotopic sites in freely moving animals. The present work was motivated by a recent study suggesting that ripple events mainly occur asynchronously between hemispheres (*Villalobos et al., 2017*), a finding that contrasts with earlier literature (*Buzsáki, 1989*; *Suzuki and Smith, 1988a*; *Suzuki and Smith, 1987*). To address this issue, we first ensured reliable ripple detection in shanks positioned in the left and right hippocampi (*Figure 1*). We found that ripple abundance, peak frequency, inter-ripple intervals, and the mean number of ripple cycles per event are equivalent between hemispheres (*Figure 2*). We next investigated if ripples occurring within the same hemisphere have different synchronization properties compared to ripples recorded in different hemispheres. Our results demonstrate: (1) high phase synchronization among ipsilateral ripples, contrasted with much lower phase-locking for contralateral ripples (*Figure 3*), (2) high amplitude correlations for ipsilateral and contralateral ripples, with slightly stronger effect for ripples within the same hemisphere (*Figure 4*), (3) larger phase-coupling differences between hemispheres compared to amplitude coupling (*Figure 5*), (4) ripple events are still highly synchronized at 5ms when detected in ipsilateral shanks, though much less so in contralateral ones; however, at broader windows of 50 and 100 ms, ripple events were highly synchronized in both ipsilateral and contralateral shanks (*Figure 6*), and, finally, (5) CA1 pyramidal neurons and interneurons exhibit local phase-locking and global time-locking to ripple events (*Figure 7*). Taken together, our results show that ripple events predominantly occur in synchrony both within and between hemispheres and not asynchronously. Our results align with previous reports (*Chrobak and Buzsáki, 1996*; *Buzsáki, 1989*; *Patel et al., 2013*; *Suzuki and Smith, 1988a*; *Buzsáki et al., 2003*) but differ from *Villalobos et al., 2017*.

The CA3 and CA1 regions form a highly interconnected structure (*Li et al., 1994*; *Laurberg, 1979*; *Amaral and Witter, 1989*; *Witter, 2007*). CA3 pyramidal neurons have highly collateralized axons that project to CA3 and CA1 via two main pathways: ipsilateral (associational) and contralateral (commissural) connection systems (*Shinohara et al., 2012*). In the associational pathway, CA3 distributes its branches along and parallel to the CA3 septo-temporal axis through the longitudinal association bundle, terminating within CA3 at the *stratum radiatum* and *stratum oriens* (*Swanson et al., 1978*; *Hjorth-Simonsen, 1973*; *Ishizuka et al., 1990*; *Li et al., 1994*). Additionally, the same CA3 pyramidal neurons give rise to another associational pathway to CA1 via the Schaffer collaterals, which target the CA1 *stratum radiatum* and *stratum oriens* (*Hjorth-Simonsen, 1973*; *Laurberg and Sørensen, 1981*; *Swanson et al., 1978*; *Swanson et al., 1980*; *Ishizuka et al., 1990*; *Witter, 2007*). The commissural system exhibits a similar organization: CA3 pyramidal neurons across the entire septo-temporal axis project to the contralateral CA3 and CA1, reaching the same target strata (*radiatum* and *oriens*) (*Laurberg, 1979*; *Laurberg and Sørensen, 1981*). Interestingly, some CA3 pyramidal neurons send their collaterals to both associational and commissural pathways (*Laurberg and Sørensen, 1981*). Finally, CA3 to CA1 excitatory projections seem unidirectional, that is, the CA1 does not project back to CA3 (*Hjorth-Simonsen, 1973*; but see *Sik et al., 1994*, for evidence of CA1 GABAergic neurons that project back to CA3).

The anatomical findings are also supported by physiological evidence. Several studies have highlighted the functional role of CA3–CA3 and CA3–CA1 excitatory connectivity. Population spikes evoked by CA3 stimulation spread across distant portions of CA1 bilaterally (*Finnerty and Jefferys, 1993*). Using electrical stimulation, *Buzsa`ki and Eidelberg, 1982* demonstrated that both CA3 associational and commissural pathways activate apical and basal dendrites of ipsilateral and contralateral CA1 pyramidal neurons. Moreover, population spikes evoked by CA3 stimulation spread across distant portions of CA1 bilaterally (*Finnerty and Jefferys, 1993*). In turn, despite the CA1 main outputs targeting the *subiculum* and entorhinal cortex, its septal (dorsal) region also projects sparsely to ipsilateral and contralateral CA1 (*Swanson et al., 1978*; *van Groen and Wyss, 1990*).

Despite the anatomical symmetry of the bilateral hippocampus, evidence for lateralization is limited (*Shinohara et al., 2008*; *Kawakami et al., 2003*; *Klur et al., 2009*; *Shipton et al., 2014*; *Villalobos et al., 2017*), and its potential impact on ripple features remains uncertain (*Villalobos et al., 2017*). Our ripple detection method followed the approach of *Villalobos et al., 2017*, allowing us to replicate similar ripple features reported in the literature. Ripple mean frequency distributions

ranged between 140 and 150 Hz (*Figure 2C, D*), consistent with findings by *Nitzan et al., 2022* and *Gan et al., 2017*. Ripple events frequently occurred as doublets, reflected by an inter-ripple interval peak between 100 ms (*Figure 2E, F*), in agreement with *Buzsáki et al., 2003*. Typically, the number of cycles per ripple event ranges from three to nine cycles (*Buzsáki, 2015*), and we found that most ripple events comprised six to seven cycles (*Figure 2G, H*), similar to *Villalobos et al., 2017*, *Ylinen et al., 1995*, and *Sullivan et al., 2011*, but fewer than reported by *Gan et al., 2017*. The ripple event rate (ripple abundance) was approximately 0.5 events per second (*Figure 2A, B*), which exceeds the rates reported by *Villalobos et al., 2017* and *Eschenko et al., 2008*, but it was lower than those reported by *Ego-Stengel and Wilson, 2010*. Most importantly, we found no significant differences in ripple features between the left and right hemispheres. Furthermore, equivalence testing between hemispheres confirmed that ripple features were indeed equivalent within the specified bounds. Thus, while some evidence suggests CA1 lateralization, it does not appear to influence the ripple features examined in this study.

The study of ripple oscillation synchrony has a long history in rodent CA1 electrophysiology; however, we believe that ripple synchrony remains underexplored, particularly from a quantitative perspective. This conclusion is based on the lack of extensive studies identified through key term searches, citation reviews, and book chapters on SWRs (*Draguhn et al., 2000*; *Andersen, 2007*; *Buzsáki, 2015*; *Liu et al., 2022*; *Morris et al., 2024*). While some authors describe SWRs as highly synchronous or among the most synchronous patterns in the mammalian brain (*Liu et al., 2022*; *Sullivan et al., 2011*; *Buzsáki, 2015*), these claims often lack clarification and are used in varying contexts, including: (1) SWRs as the result of synchronous firing from CA3 and/or CA1 neurons (*Suzuki and Smith, 1988a*; *Sullivan et al., 2011*), (2) SWRs as synchronous field events spanning the hippocampal and parahippocampal structures (*Chrobak and Buzsáki, 1994*) and (3) SWRs as simultaneous events along the hippocampal septo-temporal axis and/or bilaterally (*Suzuki and Smith, 1987*).

Another source of confusion arises from the fact that not all studies detect sharp waves and ripples simultaneously, despite referring to them collectively as SWRs. Much of the literature has focused on either sharp waves originating from the CA1 *stratum radiatum* or ripples from the CA1 *stratum pyramidale*, rather than both. Nevertheless, it is generally assumed that most sharp waves are accompanied by ripples, and vice versa. To address this ambiguity, we explicitly clarify whether a study considered only sharp waves or only ripples, as we have done in our analysis.

To our knowledge, only a few studies have investigated SWRs synchronicity between hemispheres (*Buzsáki, 1986*; *Buzsáki, 1989*; *Buzsáki et al., 2003*; *Chrobak and Buzsáki, 1996*; *Guan et al., 2021*; *Suzuki and Smith, 1987*; *Suzuki and Smith, 1988a*; *Suzuki and Smith, 1988b*; *Valeeva et al., 2019*; *Villalobos et al., 2017*), whereas intra-hemispheric ripple dynamics along both the longitudinal and transverse axes have been more extensively analyzed (*Buzsáki, 1989*; *Buzsáki et al., 1992*; *Chrobak and Buzsáki, 1996*; *Sullivan et al., 2011*; *Ylinen et al., 1995*; *Nitzan et al., 2022*; *Patel et al., 2013*; *Valeeva et al., 2020*; *Csicsvari et al., 2000*; *Villalobos et al., 2017*). Research on intra-hemispheric SWRs synchrony has demonstrated that sharp-wave and ripple events are highly phase- and/or time-locked along the longitudinal axis within the same hemisphere of rats. For example, *Buzsáki, 1989* reported the simultaneous occurrence of sharp waves and population spike bursts across distances of up to 3 mm in the rat CA1 *stratum radiatum*. Building on this, *Buzsáki et al., 1992* demonstrated ripple phase-locking across shanks spaced up to 1.8 mm along the septo-temporal axis. However, their Figure 2B reveals variability in the averaged ripple signals, without clarifying whether the diminished averaged traces are due to inter-electrode distance. *Ylinen et al., 1995* observed ripple phase-locking along the CA1 septo-temporal axis with shanks spaced 300 µm, spanning up to 1.5mm, while *Chrobak and Buzsáki, 1996* reported ipsilateral phase-locked ripples over distances of 4mn to 5mm. Afterward, *Patel et al., 2013* found that approximately 38% of ripple events were time-locked within 6 mm, 37% propagated bidirectionally along the septo-temporal axis at speeds of 0.33 to 0.4 mm/ms, and 21% were localized within 1 mm. Notably, our dataset spans 1.2 mm, and *Villalobos et al., 2017* analyzed intervals smaller than 200 µm. This suggests that propagating ripples would be separated by less than 5 ms, a time window matching the lower limit of our ripple coincidence and correlation analyses (*Figure 6*).

Sharp-wave events have been observed to occur simultaneously on ipsilateral electrodes spanning 1.2–2.4 mm in the septal and intermediate CA1 regions of neonatal rats (*Valeeva et al., 2020*). Furthermore, *Nitzan et al., 2022* reported that ripple features such as amplitude, frequency, and duration

remain consistent along the longitudinal axis of CA1, with ripple events time-locked across the entire hippocampal axis, spanning from the CA1 septal regions to posterior areas and the *subiculum* (3 mm in mice). In contrast, *Csicsvari et al., 2000* and *Sullivan et al., 2011* investigated ripple synchrony along the mediolateral axis. *Sullivan et al., 2011* found high ripple amplitude correlations along the CA1 transverse axis in rats using shanks spaced 300 μm, while *Csicsvari et al., 2000* reported a high ripple coincidence probability within 50 ms bins using electrodes with similar spacing. Collectively, these studies demonstrate that sharp-wave events and ripple oscillations are highly phase- and/or time-locked along both the septo-temporal and transverse axes within the same CA1 hemisphere.

Conversely, inter-hemispheric ripple synchrony appears less phase-locked but still exhibits temporal coordination in the rat CA1. *Buzsáki, 1986* demonstrated simultaneous sharp waves in contralateral CA1 *stratum radiatum* spaced 5 mm apart, with a time lag of less than 5ms and similar amplitudes triggered by homotopic regions. Similarly, *Suzuki and Smith, 1987* presented a bilateral recording example of synchronous sharp waves between electrodes spaced 5 mm (homotopic sites). They also noted that sharp waves occurred simultaneously in both homotopic and heterotopic CA1 sites, with spatial shifts of 1–2.5 mm; however, this early study lacked a thorough quantification of synchronization. In a subsequent study, *Suzuki and Smith, 1988a* adopted a more quantitative approach, showing several superimposed traces of bilateral sharp waves spaced 5mm (homotopic sites). They suggested that contralateral ripples are not strongly phase-locked but may appear either in phase or out of phase. *Buzsáki, 1989* provided an example of simultaneous bilateral sharp waves in rat CA1, also spaced approximately 5mm (based on our estimates). *Chrobak and Buzsáki, 1996* reported increased ripple event coincidences between contralateral electrodes spaced 3mm, although phase coupling remained restricted to ipsilateral recordings. *Buzsáki et al., 2003* observed a diminished ripple-filtered average response when using a contralateral ripple cycle as a trigger (similar to our *Figure 3E, F*). Nevertheless, ripple amplitudes were correlated, indicating contralateral synchrony in timing but not in phase (also similar to our *Figure 4*). Studies on neonatal rats by *Valeeva et al., 2019* demonstrated a strong amplitude correlation and high coincidence of sharp-wave events between homotopic contralateral CA1 sites spaced up to approximately 2.4 mm. Interestingly, *Guan et al., 2021* showed that silencing CA3 reduces inter-hemispheric ripple event cross-correlation, suggesting that CA3 provides a common bilateral input governing CA1 ripple timing. Collectively, these findings, along with ours, suggest that global ripple synchronization depends more on event timing than on phase coupling. In ipsilateral recordings, distance plays a critical role in phase-locking, with phase coupling strength exhibiting a linear decline as electrode spacing increases along the septo-temporal axis of CA1.

A recent study by *Villalobos et al., 2017* challenged the notion that hippocampal ripple oscillations occur simultaneously across both hemispheres. To test this, they employed an arrangement of eight tetrodes, four in each hemisphere. According to their diagram, ipsilateral electrodes were spaced less than 200 μm, while contralateral electrodes were separated by up to 4.5mm. Due to their random electrode selection, ipsilateral electrodes could be positioned along either the medio-lateral or septo-temporal axis. By detecting ripple events from ipsilateral and contralateral hemispheres, they found that most contralateral ripples occurred independently. Specifically, they reported an ipsilateral coincidence of 22% within 5ms time bins and 29% within 100 ms bins, compared to a contralateral coincidence of only 3% for 5ms bins and 10% for 100 ms bins. This result is striking not only because of the low coincidence of ripples between contralateral sites but also due to the relatively low coincidence of ripples between electrodes spaced ≈ 125 μm apart within the same hemisphere.

Finally, we observed an increase in overall firing rates during ripple events, with CA1 pyramidal neurons firing maximally just before the ripple trough and interneuron activity peaking just afterward. This finding aligns with previous studies (*Buzsáki, 1986*; *Buzsáki et al., 1992*; *Chrobak and Buzsáki, 1996*; *Csicsvari et al., 1999a*; *Csicsvari et al., 1999b*; *Csicsvari et al., 2000*; *Buzsáki et al., 2003*). Our data further demonstrate that pyramidal neurons exhibit local ripple phase coupling, with coupling strength decaying rapidly in both ipsilateral and contralateral directions. In contrast, interneurons maintain a diminished but significant phase coupling to contralateral ripples, likely supported by their connections to commissural and associational excitatory fibers (*Deller et al., 1994*). Supporting this, *Stark et al., 2014* showed that localized CA1 pyramidal activation induces ripple events with multiple ipsilateral loci phase-locking ripple cycles. They also demonstrated that disrupting GABA$_A$ signaling or silencing PV+ basket cells reduces spatial and spiking coherence in the ripple band while increasing

interneuron phase-locking. Similarly, *Patel et al., 2013* reported that the phase-locking between ripples and multi-unit spiking activity diminishes with increasing distances from ripples detected at the most septal CA1 recording site but remains consistently time-locked to them. Additionally, *Suzuki and Smith, 1988b* found that high-frequency stimulation of anterior CA1 evokes ripple-like events bilaterally, likely mediated by CA3 bursts, consistent with our observation of a global increase in firing rates during ripples. Firing rate synchronization during ripples also occurs along the medio-lateral axis. *Sullivan et al., 2011* and *Csicsvari et al., 2000* showed that interneurons couple to distant recording sites along the transverse axis, whereas pyramidal cells remain locally coupled – a pattern we also observed along the longitudinal axis. One potential role of time-locked contralateral ripples could be to synchronize place cells across hemispheres, facilitating the formation of global cell assemblies (*Pfeiffer and Foster, 2015*).

## Conclusions and future directions

In summary, despite the long history of studies on CA1 ripple oscillations, our work provides one of the few detailed quantitative analyses of their inter-hemispheric synchronization. Notably, our findings challenge the conclusions of *Villalobos et al., 2017* by demonstrating that ripple events, while primarily phase-locked within the same hemisphere, exhibit robust time-locking across hemispheres. This indicates that ripple synchronization is not purely local but involves global temporal coordination, likely driven by common bilateral inputs from CA3.

Our results further reveal that phase and amplitude coupling metrics capture distinct aspects of ripple synchrony. Phase coupling is highly localized, rapidly decaying with distance along the septo-temporal axis, whereas amplitude coupling remains consistent over broader spatial scales, highlighting the global nature of ripple time-locking. These findings suggest that the hippocampus balances localized processing with global coordination, a mechanism that may support memory consolidation and the formation of coherent cell assemblies across hemispheres.

Although our results robustly demonstrate the time-locked co-occurrence of ripples across hemispheres, they do not directly address whether the information encoded by each hemisphere is correlated. One promising avenue would be to investigate neuronal assembly dynamics during exposure to novelty, given that awake SWR-associated replay can express both forward and reverse reactivation of behavioral sequences, even in unfamiliar environments (*Buhry et al., 2011*). While such analyses lie beyond the scope of the present study, they represent a clear and compelling direction for future work.

# Materials and methods

**Key resources table**

| Reagent type (species) or resource | Designation | Source or reference | Identifiers | Additional information |
|---|---|---|---|---|
| Software, algorithm | Python 3.12 | python.org | RRID:SCR_008394 | Programming language used in analysis |
| Software, algorithm | NumPy | scipy.org | RRID:SCR_008633 | Fundamental numerical library |
| Software, algorithm | SciPy | numpy.org | RRID:SCR_008058 | Scientific computing library |
| Software, algorithm | Pandas | pandas.pydata.org | RRID:SCR_018214 | Data manipulation library |
| Software, algorithm | Matplotlib | matplotlib.org | RRID:SCR_008624 | Plotting library |
| Software, algorithm | Seaborn | seaborn.pydata.org | RRID:SCR_018132 | Statistical data visualization built on matplotlib |
| Software, algorithm | Statsmodels | statsmodels.org | RRID:SCR_016074 | Statistical modeling and tests library |
| Software, algorithm | Pingouin | pingouin-stats.org | RRID:SCR_022261 | Statistical analysis package |

*Continued on next page*

*Continued*

| Reagent type (species) or resource | Designation | Source or reference | Identifiers | Additional information |
|---|---|---|---|---|
| Software, algorithm | DABEST | acclab.github.io | RRID:SCR_022340 | Python package for estimation statistics focusing on effect sizes |
| Software, algorithm | Inkscape | inkscape.org | RRID:SCR_014479 | Vector graphics editor used for figure editing |
| Software, algorithm | Code repository | https://github.com/RobsonSchefferTeixeira/elife_ripple_synchronization, *Teixeira, 2025* | | Analysis code and scripts made available on author's GitHub |

## Datasets

We analyzed a dataset of four male Long Evans rats from the Buzsáki Lab (*Grosmark and Buzsáki, 2016*; *Chen et al., 2016*) generously made available online at https://crcns.org/ (*Grosmark et al., 2016*). The animals were implanted with two 6- or 8-shank silicon probes of 10 or 6 vertically aligned contacts, respectively. Inter-shank distance was 200 µm and inter-electrode distance within shanks was 20 µm. The shanks were aimed at the dorsal CA1 pyramidal layer along the septo-temporal axis and bilaterally implanted. For each shank, we selected the channel with the highest mean ripple power (*Figure 1A*). Although the recordings did not specifically target the *stratum radiatum* of CA1, sharp negative deflections associated with ripples were visible in electrodes located at the inferior border of *stratum pyramidale* near or at the beginning of the *stratum radiatum* and were also used to assess ripple detection accuracy. At this step, for each shank, we selected the electrode with the highest ripple amplitude, corresponding to the center of the dorsal CA1 pyramidal layer (*Mizuseki et al., 2011*). Shanks containing high-frequency artifacts that lacked a clear ripple pattern were excluded.

To validate our results, we analyzed an independent publicly available dataset from the Zugaro Lab, accessible at https://crcns.org/ (*Drieu et al., 2018a*; *Drieu et al., 2018b*). This dataset contains bilateral recordings from the dorsal CA1 pyramidal layer of five Long-Evans rats during baseline sleep. Recordings were made using 16 tetrodes or octrodes (eight per hemisphere) targeted at CA1 *stratum pyramidale*. Following the approach in *Grosmark et al., 2016*, we selected one electrode per tetrode/octrode exhibiting the highest ripple amplitude and excluded electrodes contaminated by artifacts or lacking ripple events.

## Analyses

We performed all analyses using Python 3.12 and widely used libraries, including numpy, scipy, pandas, seaborn, matplotlib, and DABEST. We also designed linear models for statistical analyses using the statsmodels and pingouin Python packages.

## Epochs

In *Grosmark et al., 2016* dataset, animals had their sleep–wake cycles (≈4 h) recorded during each session, both before and after exposure to a novel environment (linear and circular tracks, where the animals ran through the maze to obtain a water reward). The same animals experienced each maze only once. To avoid muscular artifacts and maximize the detection of ripple events, we focused our main analysis on 1h of concatenated baseline SWS epochs, taken from the pre-maze sleep periods recorded before exposure to the novel environment. These epochs were characterized by a high delta–theta ratio, absence of movements, presence of irregular slow frequency activity, and SWR events in CA1 (*Figure 1*). This epoch length matches that used by *Villalobos et al., 2017* and, similarly, includes only baseline sleep, thus avoiding novelty-induced changes in ripple statistics. To assess the effect of novelty exposure, we additionally analyzed 30min of SWS epochs recorded both before and after maze sessions, following the approach used in *Eschenko et al., 2008*.

For the *Drieu et al., 2018a* dataset, we used ≈1 h of concatenated SWS epochs from baseline sleep. These replication results are shown in *Figure 4—figure supplement 2*.

## Ripple detection and thresholds

We detected ripple events by first filtering the LFP signal between 100–250 Hz using the filrs function from the scipy toolkit. This function applies a non-causal filter, processing the signal in both forward and reverse directions to avoid phase distortions. The filtered signal was then Hilbert-transformed, and the absolute value of the analytic signal was calculated to extract the instantaneous amplitude envelope. Following the method described in *Villalobos et al., 2017*, we classified ripple events as those with a local maximum exceeding 2 standard deviations above the mean ripple amplitude, and a minimum duration of $30\,\mathrm{ms}$ around the peak amplitude. The inter-ripple interval was calculated as the time difference between two consecutive ripple peak events.

## Wavelet and ripple frequency estimation

For the time-frequency analyses depicted in *Figures 1 and 2*, we convoluted the LFP signal with a set of complex Morlet wavelets. Each wavelet was constructed with a center frequency linearly increasing from 100 to 250 Hz (2 Hz steps), all containing six cycles. We then obtained time–frequency plots by using the instantaneous amplitude envelope of each convolution. Next, we constructed a time–frequency decomposition window around each ripple event and calculated the mean amplitude for each frequency across the event. The frequency with the maximum amplitude was then considered the peak frequency of the ripple event (*Figures 1C and 2C, D*).

## Phase-locking value

To assess PPC between ripples recorded at ipsilateral and contralateral electrodes, we (1) obtained the ripple-filtered time series, (2) Hilbert-transformed the filtered signal and generated the analytical signal, and (3) extracted the instantaneous phase series, in radians, using the arctangent function. We used the PLV as an index of phase synchronization between two ripple-filtered series (*Lachaux et al., 1999*). The PLV is computed as the length of the mean resultant vector over unitary phase-difference vectors. When phase differences are randomly distributed across the circle, the resultant length approaches zero (i.e., the unitary vectors cancel each other); conversely, if phase differences remain constant, the resultant length is one (i.e., the mean of perfectly aligned unitary vectors).

The formula for the PLV between two phase series is given by

$$\mathrm{PLV} = \left| \frac{1}{N} \sum_{n=1}^{N} e^{i\Delta\phi_n} \right|$$

where

$$\Delta\phi_n = \phi_{1,n} - \phi_{2,n}$$

- $\phi_{1,n}$ and $\phi_{2,n}$ represent the instantaneous phases of two ripple-filtered signals at time point $n$;
- $\Delta\phi_n$ is the phase difference between the two ripple-filtered signals;
- $N$ is the total number of time points;
- $e^{i\Delta\phi_n}$ represents the unitary phase difference vector in the complex plane;
- $|\cdot|$ denotes the magnitude of the mean resultant vector.

## Amplitude correlation

To assess AAC among shanks, we first extracted the instantaneous amplitude series of the ripple-filtered LFP as described above. Then, we computed the mean amplitude over sliding windows of 50 ms with 10 ms overlap. Finally, we used a linear regression model to calculate the correlation coefficient between shanks: a coefficient near one indicates high amplitude coupling, while a coefficient near zero reflects the absence of coupling (*Scheffer-Teixeira and Tort, 2022*).

The correlation coefficient is calculated as

$$r = \frac{\sum_{n=1}^{N}(A_{1,n} - \overline{A}_1)(A_{2,n} - \overline{A}_2)}{\sqrt{\sum_{n=1}^{N}(A_{1,n} - \overline{A}_1)^2 \sum_{n=1}^{N}(A_{2,n} - \overline{A}_2)^2}}$$

where

- $A_{1,n}$ and $A_{2,n}$ are the mean amplitudes of the ripple-filtered signals from two shanks at time point $n$, computed over sliding windows of 50 ms with 10 ms overlap.
- $\overline{A}_1$ and $\overline{A}_2$ are the mean amplitudes of the two signals over all time windows.
- $N$ is the total number of time windows.

## Cross-correlations

We computed standard cross-correlations employing either ripple-filtered signals, instantaneous ripple amplitude, or binned ripple events (5, 50, or 100 ms bin widths) from ipsilateral and contralateral shanks. The cross-correlation value at zero lag was used as a measure of synchrony. Notice that the cross-correlation between ripple-filtered signals reflects PPC, while the cross-correlation between instantaneous amplitudes reflects AAC.

## Ripple coincidence

We defined ripple coincidences as instances where the timestamps of maximum amplitude for ripple events on different shanks were separated by less than a given time window. To align with previous work, we used the bin sizes reported by *Villalobos et al., 2017* (5 and 100 ms) and by *Csicsvari et al., 2000* (ranging from 5 to 50 ms). Because coincidence proportion is a monotonically non-decreasing function of window size, a sufficiently large window can artificially inflate coincidence values. Therefore, we included an intermediate bin of 50 ms to show that it already approaches the value obtained with 100 ms, indicating that this upper limit does not substantially overestimate ripple coincidence (*Figure 6*).

In order to quantify coincidence, we first detected ripple events on a reference shank and calculated the proportion of these events coinciding with ripple events on another shank (ipsilateral or contralateral). The process was then repeated using the other shank as the reference. The final proportion was obtained by averaging the results from both reference configurations. These findings are presented as the mean proportion of coincident ripple events relative to the total number of detected ripple events.

## Spiking analyses

The dataset from Buzsáki Lab (*Grosmark and Buzsáki, 2016*) contained spike-sorted neurons classified into pyramidal neurons ($N_{\mathrm{pyr}} = 559$) and interneurons ($N_{\mathrm{int}} = 128$). We examined the relationship between spiking activity and ripple events using similar metrics as in ripple signal analyses. For phase coupling, we applied the PLV method, using the ripple phase during which spikes occurred. And since amplitude coupling primarily quantifies the timing between ripple events, we used the firing rate difference between inside and outside of ripple events as a time-locking index.

To assess the significance of phase coupling, as shown in *Figure 7I, J* reference distribution of PLV values was created by randomly shifting spike times by ±10 s, repeating the process 300 times, and calculating the PLV for each shift. The p-value was determined as the proportion of shifted PLVs from the control distribution that exceeded the original PLV. A p-value < 0.05 indicated significant phase coupling between spikes and ripple phase.

To calculate the PLV for spike-ripple phase coupling, we used the following formula:

$$\mathrm{PLV} = \left| \frac{1}{N_{\mathrm{spikes}}} \sum_{k=1}^{N_{\mathrm{spikes}}} e^{i\phi_k} \right|$$

where

- $N_{\mathrm{spikes}}$ is the total number of spikes;
- $\phi_k$ is the ripple phase at the time of the $k$th spike;
- $e^{i\phi_k}$ represents the unitary phase vector corresponding to the phase of each spike in the ripple cycle;
- $|\cdot|$ denotes the magnitude of the mean resultant vector, quantifying the consistency of spike phases relative to the ripple cycle.

## Statistics

### Data structure

We designed an analysis scheme to study all combinations of ipsilateral and contralateral shanks (see *Figure 1D*). Each shank, which provided a single electrode, was positioned along the septo-temporal axis of the dorsal CA1. For ipsilateral recordings, comparing shank 1 with shank 2 or shank 3 with shank 4 resulted in a relative distance of 200 µm in both cases. For contralateral recordings, we applied the same ordering logic to allocate distances, ensuring they reflected the relative positions from the homotopic site rather than absolute lateral distances, which exceed 2000 µm. Thus, left shank 1 and right shank 1 were assigned a 0 µm distance, as they occupy the same relative position in the contralateral homotopic site. Likewise, left shank 3 and right shank 4 were assigned a 200 µm distance. This approach aligns with evidence showing that CA3–CA1 contralateral projections are denser for homotopic sites (*Laurberg, 1979*).

### Descriptive statistics

To visualize the distribution and variability of the data, we employed a modified version of Raincloud plots (*Allen et al., 2019*). These plots combine a half-violin plot to represent data density with a swarm plot, displaying individual data points. A boxplot is also included, but without outliers, as they are redundant when shown alongside the swarm plot. In graphs featuring only boxplots, individual dots represent outliers.

### Model

We employed an LMMR from statsmodels to estimate the effect of a dependent variable while accounting for independent variables (animals), nested variables (sessions within animals), and repeated measures (shanks recorded within a single session). Detailed test results can be found in the *Supplementary file 1*. The model also included a random intercept for each rat identity to account for potential correlations between observations within the same rat due to shared characteristics. Specifically, the LMMR estimates the coefficients for independent variables, which represent the mean difference between the reference level (*left hippocampus* for *Figure 2* and *ipsilateral shanks* for *Figures 3–7*) and the remaining levels (*right hippocampus* and *contralateral shanks*, respectively). Statistical significance was assessed using a *z*-test between groups. Positive coefficients indicate an increase on the *right side* or *contralateral* relative to the reference, while negative coefficients indicate a decrease.

The relative percentage difference between contra and ipsi is calculated as

$$\left| \frac{\text{contra} - \text{ipsi}}{\text{ipsi}} \right| \times 100\%$$

For a model with only the *side* variable, this translates to

$$\left| \frac{\text{coefficient}}{\text{intercept}} \right| \times 100\%$$

And for a linear regression model:

$$\left| \frac{\text{slope}}{\text{intercept}} \right| \times 100\%$$

Negative percentages indicate a reduction in the relative percentage difference.

When studying the outcome effect of shank distance (*Figures 3D, G and 4D, G*, and *Figure 7—figure supplement 2*), we first employed a model that included all variables: *side*, *shank distance*, and *interaction*. If an interaction effect was observed, we analyzed the distance across each side independently, as the data suggested that the correlation line slope differed for each *side* level.

We deemed p-values < 0.05 as significant (α or Type 1 error at 5%). We report the exact p-values across the text for transparency; in cases where p-values were extremely small (sometimes virtually zero), we represent them p-value ≈ 0. However, readers are cautioned not to interpret p-values as direct measures of evidence strength (*Lakens, 2022*).

## Effect size

To move beyond dichotomous decisions based solely on p-values, we utilized estimation statistics via the DABEST Python package (Data Analysis with Bootstrapped ESTimation; *Ho et al., 2019*). Estimation statistics advance classical significance testing by focusing on effect sizes and their associated confidence intervals. Specifically, we used Hedges' $g$, an unbiased estimator of Cohen's $d$, as the measure of effect size; 95% confidence intervals were calculated using a bootstrapped distribution with 5000 resamples. Additionally, p-values corresponding to Hedges' $g$ were derived from a two-sided permutation $t$-test, performed with 5000 reshuffles of the control and test labels under the null hypothesis of zero difference.

## Test of equivalence

In *Figure 2A, C, E, G*, we have not found any statistically significant result when comparing ripple features from left and right hippocampus, which argues against specialized hemispheres. However, rejecting the null hypothesis alone is insufficient to claim that both sides display the same ripple features, as the absence of evidence is not evidence of absence. To provide evidence that left and right hippocampus do not have specialized ripple features, we employed a test for equivalence via the TOST procedure (*Lakens, 2017*).

The TOST procedure is a statistical method used to test for equivalence rather than difference. Unlike traditional hypothesis tests that assess whether two means differ, TOST evaluates whether the difference between two means lies within a specified equivalence range set by positive and negative bounds. In this method, two one-sided composite null hypotheses are tested:

Lower bound test

$$H_0 : \mu_1 - \mu_2 \leq -\Delta_L \quad vs. \quad H_1 : \mu_1 - \mu_2 > -\Delta_L$$

Upper bound test

$$H_0 : \mu_1 - \mu_2 \geq \Delta_U \quad vs. \quad H_1 : \mu_1 - \mu_2 < \Delta_U$$

where

- $\mu_1$ and $\mu_2$ are the groups means (e.g., left or right);
- $\Delta_L$ and $\Delta_U$ are the equivalence bounds (in the same unit as the groups).

Decision rule

$$\text{Conclude equivalence if: } p_{\text{lower}} < \alpha \quad \text{and} \quad p_{\text{upper}} < \alpha$$

If equivalence is achieved, we conclude that

$$-\Delta_L < \mu_1 - \mu_2 < \Delta_U$$

This indicates that the observed effect lies within the established equivalence bounds, and therefore, the two distributions are deemed equivalent under the specified bounds.

We based the equivalence test bounds on the minimum detectable effect (MDE), that is, the smallest true effect size reliably detectable with a given level of statistical power $(1 - \beta)$, significance level $(\alpha)$ and sample size. In this study, we set the Type 1 error rate $\alpha$ = 5% and Type 2 error rate $\beta$ = 20% (i.e., power = 80 μm), and obtained an MDE of 0.6 (Cohen's $d$). Therefore, our equivalence bounds were $d$ = –0.6 to 0.6, with effect sizes below this threshold considered too small to be of interest. Additionally, we transformed the Cohen' $d = \pm 0.6$ back to the units of the analyzed feature. TOST results are included in the *Supplementary file 1*.

## Acknowledgements

Supported by Conselho Nacional de Desenvolvimento Científico e Tecnológico (CNPq) and Coordenação de Aperfeiçoamento de Pessoal de Nível Superior (CAPES). The authors thank the Buzsáki lab and the Zugaro lab for making data publicly available at crcns.org, a data-sharing website supported by NSF and NIH, USA.

## Additional information

### Funding

| Funder | Grant reference number | Author |
|---|---|---|
| Coordenação de Aperfeiçoamento de Pessoal de Nível Superior | | Robson Scheffer-Teixeira Adriano BL Tort |
| Conselho Nacional de Desenvolvimento Científico e Tecnológico | | Robson Scheffer-Teixeira Adriano BL Tort |

The funders had no role in study design, data collection, and interpretation, or the decision to submit the work for publication.

### Author contributions

Robson Scheffer-Teixeira, Conceptualization, Data curation, Software, Formal analysis, Validation, Investigation, Visualization, Methodology, Writing – original draft, Writing – review and editing; Adriano BL Tort, Conceptualization, Supervision, Funding acquisition, Visualization, Writing – original draft, Project administration, Writing – review and editing

### Author ORCIDs

Robson Scheffer-Teixeira ⓘ https://orcid.org/0009-0001-2896-742X
Adriano BL Tort ⓘ https://orcid.org/0000-0002-9877-7816

Reviewer #2 (Public review): https://doi.org/10.7554/eLife.106201.3.sa1
Author response https://doi.org/10.7554/eLife.106201.3.sa2

## Additional files

### Supplementary files

MDAR checklist

Supplementary file 1. Statistical table summarising all statistical comparisons carried out in the study.

### Data availability

All datasets are publicly available through CRCNS.org, a data-sharing repository supported by the U.S. National Science Foundation (NSF) and National Institutes of Health (NIH): Buzsaki lab: https://crcns.org/data-sets/hc/hc-11/about-hc-11; Zugaro lab: https://crcns.org/data-sets/hc/hc-18/about-hc-18.

The following previously published datasets were used:

| Author(s) | Year | Dataset title | Dataset URL | Database and Identifier |
|---|---|---|---|---|
| Grosmark AD, Long J, Buzsáki G | 2016 | Recordings from hippocampal area CA1, PRE, during and POST novel spatial learning | https://doi.org/10.6080/K0862DC5 | Collaborative Research in Computational Neuroscience, 10.6080/K0862DC5 |
| Drieu C, Todorova R, Zugaro M | 2018 | Bilateral recordings from dorsal hippocampal area CA1 from rats transported on a model train and sleeping | https://doi.org/10.6080/K0Z899MM | Collaborative Research in Computational Neuroscience, 10.6080/K0Z899MM |

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
