## [Editor Report · eLife Assessment]

This **important** study provides new insights into the synchronization of ripple oscillations in the hippocampus, both within and across hemispheres. Using carefully designed statistical methods, it presents **compelling** evidence that synchrony is significantly higher within a hemisphere than across. This study will be of interest to neuroscientists studying the hippocampus and memory.

---

## [Referee Report · Reviewer #2 (Public review)]

Summary

The authors completed a statistically rigorous analysis of the synchronization of sharp-wave ripples in the hippocampal CA1 across and within hemispheres. They used a publicly available dataset (collected in the Buzsaki lab) from 4 rats (8 sessions) recorded with silicon probes in both hemispheres. Each session contained approximately 8 hours of activity recorded during rest. The authors found that the characteristics of ripples did not differ between hemispheres, and that most ripples occurred almost simultaneously on all probe shanks within a hemisphere as well as across hemispheres. The differences in amplitude and exact timing of ripples between recording sites increased slightly with distance between recording sites. However, the phase coupling of ripples (in the 100-250 Hz range), changed dramatically with distance between recording sites. Ripples in opposite hemispheres were about 90% less coupled than ripples on nearby tetrodes in the same hemisphere. Phase coupling also decreased with distance within the hemisphere. Finally, pyramidal cell and interneuron spikes were coupled to the local ripple phase and less so to ripples at distant sites or the opposite hemisphere.

The authors also analyzed the changes in ripple coupling in relation to a couple of behavioral variables. Interestingly, while exposure to a novel track increased ripple abundance by ~5%, it did not change any form of ripple coupling within or between hemispheres.

Strengths

The analysis was well-designed and rigorous. The authors used statistical tests well suited to the hypotheses being tested, and clearly explained these tests. The paper is very clearly written, making it easy to understand and reproduce the analysis. The authors included an excellent review of the literature to explain the motivation for their study.

Weaknesses

The authors have addressed all of my concerns and recommendations.

This paper presents an important and unique analysis of ripple coupling. The same method could be used in the future to analyze the effects of other behavioral variables, such as satiety versus hunger, sleep deprivation, or enrichment, to address potential functions and causes of ripple coupling.

---

## [Author Response]

The following is the authors’ response to the original reviews.

**Reviewer #1 (Public review):**
Summary:In this manuscript, the authors analyze electrophysiological data recorded bilaterally from the rat hippocampus to investigate the coupling of ripple oscillations across the hemispheres. Commensurate with the majority of previous research, the authors report that ripples tend to co-occur across both hemispheres. Specifically, the amplitude of ripples across hemispheres is correlated but their phase is not. These data corroborate existing models of ripple generation suggesting that CA3 inputs (coordinated across hemispheres via the commisural fibers) drive the sharp-wave component while the individual ripple waves are the result of local interactions between pyramidal cells and interneurons in CA1.Strengths:The manuscript is well-written, the analyses well-executed and the claims are supported by the data.Weaknesses:One question left unanswered by this study is whether information encoded by the right and left hippocampi is correlated.

Thank you for raising this important point. While our study demonstrates ripple co-occurrence across hemispheres, we did not directly assess whether the information encoded in each hippocampus is correlated. Addressing this question would require analyses of coordinated activity patterns, such as neuronal assemblies formed during novelty exposure, which falls beyond the scope of the present study. However, we agree this is an important avenue for future work, and we now acknowledge this limitation and outlined it as a future direction in the Conclusion section (lines 796–802).

**Reviewer #2 (Public review):**
Summary:The authors completed a statistically rigorous analysis of the synchronization of sharp-wave ripples in the hippocampal CA1 across and within hemispheres. They used a publicly available dataset (collected in the Buzsaki lab) from 4 rats (8 sessions) recorded with silicon probes in both hemispheres. Each session contained approximately 8 hours of activity recorded during rest. The authors found that the characteristics of ripples did not differ between hemispheres, and that most ripples occurred almost simultaneously on all probe shanks within a hemisphere as well as across hemispheres. The differences in amplitude and exact timing of ripples between recording sites increased slightly with the distance between recording sites. However, the phase coupling of ripples (in the 100-250 Hz range), changed dramatically with the distance between recording sites. Ripples in opposite hemispheres were about 90% less coupled than ripples on nearby tetrodes in the same hemisphere. Phase coupling also decreased with distance within the hemisphere. Finally, pyramidal cell and interneuron spikes were coupled to the local ripple phase and less so to ripples at distant sites or the opposite hemisphere.Strengths:The analysis was well-designed and rigorous. The authors used statistical tests well suited to the hypotheses being tested, and clearly explained these tests. The paper is very clearly written, making it easy to understand and reproduce the analysis. The authors included an excellent review of the literature to explain the motivation for their study.Weaknesses:The authors state that their findings (highly coincident ripples between hemispheres), contradict other findings in the literature (in particular the study by Villalobos, Maldonado, and Valdes, 2017), but fail to explain why this large difference exists. They seem to imply that the previous study was flawed, without examining the differences between the studies.The paper fails to mention the context in which the data was collected (the behavior the animals performed before and after the analyzed data), which may in fact have a large impact on the results and explain the differences between the current study and that by Villalobos et al. The Buzsaki lab data includes mice running laps in a novel environment in the middle of two rest sessions. Given that ripple occurrence is influenced by behavior, and that the neurons spiking during ripples are highly related to the prior behavioral task, it is likely that exposure to novelty changed the statistics of ripples. Thus, the authors should analyze the pre-behavior rest and post-behavior rest sessions separately. The Villalobos et al. data, in contrast, was collected without any intervening behavioral task or novelty (to my knowledge). Therefore, I predict that the opposing results are a result of the difference in recent experiences of the studied rats, and can actually give us insight into the memory function of ripples.

We appreciate this thoughtful hypothesis and have now addressed it explicitly. Our main analysis was conducted on 1-hour concatenated SWS epochs recorded before any novel environment exposure (baseline sleep). This was not clearly stated in the original manuscript, so we have now added a clarifying paragraph (lines 131–143). The main findings therefore remain unchanged.

To directly test the reviewer’s hypothesis, we performed the suggested comparison between pre- and post-maze rest sessions, including maze-type as a factor. These new analyses are now presented in a dedicated Results subsection (lines 475 - 493) and in Supplementary Figure 5.1. While we observed a modest increase in ripple abundance after the maze sessions — consistent with known experienced-dependent changes in ripple occurrence — the key findings of interhemispheric synchrony remained unchanged. Both pre- and post-maze sleep sessions showed robust bilateral time-locking of ripple events and similar dissociations between phase and amplitude coupling across hemispheres.

In one figure (5), the authors show data separated by session, rather than pooled. They should do this for other figures as well. There is a wide spread between sessions, which further suggests that the results are not as widely applicable as the authors seem to think. Do the sessions with small differences between phase coupling and amplitude coupling have low inter-hemispheric amplitude coupling, or high phase coupling? What is the difference between the sessions with low and high differences in phase vs. amplitude coupling? I noticed that the Buzsaki dataset contains data from rats running either on linear tracks (back and forth), or on circular tracks (unidirectionally). This could create a difference in inter-hemisphere coupling, because rats running on linear tracks would have the same sensory inputs to both hemispheres (when running in opposite directions), while rats running on a circular track would have different sensory inputs coming from the right and left (one side would include stimuli in the middle of the track, and the other would include closer views of the walls of the room). The synchronization between hemispheres might be impacted by how much overlap there was in sensory stimuli processed during the behavior epoch.

Thank you for this insightful suggestion. In our new analyses comparing pre- and post-maze sessions, we have also addressed this question. Supplementary Figures 4.1 and 5.1 (E-F) present coupling metrics averaged per session and include coding for maze type. Additionally, we have incorporated the reviewer’s hypothesis regarding sensory input differences and their potential impact on inter-hemispheric synchronization into a new Results subsection (lines 475–493).

The paper would be a lot stronger if the authors analyzed some of the differences between datasets, sessions, and epochs based on the task design, and wrote more about these issues. There may be more publicly available bi-hemispheric datasets to validate their results.

To further validate our findings, we have analyzed another publicly available dataset that includes bilateral CA1 recordings (https://crcns.org/data-sets/hc/hc-18). We have added a description of this dataset and our analysis approach in the Methods section (lines 119–125 and 144-145), and present the corresponding results in a new Supplementary Figure (Supplementary Figure 4.2). These new analyses replicated our main findings, confirming robust interhemispheric time-locking of ripple events and a greater dissociation between phase and amplitude coupling in ipsilateral versus contralateral recordings.

**Reviewer #1 (Recommendations for the authors):**
My only suggestion is that the introduction can be shortened. The authors discuss in great length literature linking ripples and memory, although the findings in the paper are not linked to memory. In addition, ripples have been implicated in non-mnemonic functions such as sleep and metabolic homeostasis.

The reviewer`s suggestion is valid and aligns with the main message of our paper. However, we believe that the relationship between ripples and memory has been extensively discussed in the literature, sometimes overshadowing other important functional roles (based on the reviewer’s comment, we now also refer to non-mnemonic functions of ripples in the revised introduction [lines 87–89]). Thus, we find it important to retain this context because highlighting the publication bias towards mnemonic interpretations helps frame the need for studies like ours that revisit still incompletely understood basic ripple mechanisms.

We also note that, based on a suggestion from reviewer 2, we have supplemented our manuscript with a new figure demonstrating ripple abundance increases during SWS following novel environment exposure (Supplementary Figure 5.1), linking it to memory and replicating the findings of Eschenko et al. (2008), though we present this result as a covariate, aimed at controlling for potential sources of variation in ripple synchronization.

**Reviewer #2 (Recommendations for the authors):**
It would be useful to include more information about the analyzed dataset in the methods section, e.g. how long were the recordings, how many datasets per rat, did the authors analyze the entire recording epoch or sub-divide it in any way, how many ripples were detected per recording (approximately).

We have now included more detailed information in the Methods section (lines 104 - 145).

A few of the references to sub-figures are mislabeled (e.g. lines 327-328).

Thank you for noticing these inconsistencies. We have carefully reviewed and corrected all figure sub-panel labels and references throughout the manuscript.

In Figure 7 C&D, are the neurons on the left sorted by contralateral ripple phase? It doesn't look like it. It would be easier to compare to ipsilateral if they were.

In Figures 7C and 7D, neurons are sorted by their ipsilateral peak ripple phase, with the contralateral data plotted using the same ordering to facilitate comparison. To avoid confusion, we have clarified this explicitly in the figure legend and corresponding main text (lines 544–550).

In Figure 6, using both bin sizes 50 and 100 doesn't contribute much.

We used both 50 ms and 100 ms bin sizes to directly compare with previous studies (Villalobos et al. 2017 used 5 ms and 100 ms; Csicsvari et al. 2000 used 5–50 ms). Because the proportion of coincident ripples is a non-decreasing function of the window size, larger bins can inflate coincidence measures. Including a mid-range bin of 50 ms allowed us to show that high coincidence levels are reached well before the 100 ms upper bound, supporting that the 100 ms window is not an overshoot. We have added clarification on this point in the Methods section on ripple coincidence (lines 204–212).